

# Knickpoints and Fixpoints: The Evolution of Fluvial Morphology under the Combined Effect of Fault Uplift and Dam Obstruction on a Soft Bedrock River

Hung-En Chen[1], Yen-Yu Chiu[1], Chih-Yuan Cheng[1] and Su-Chin Chen[1,2]

[1] Department of Soil and Water Conservation, National Chung Hsing University, Taichung 40227, Taiwan

[2] Innovation and Development Center of Sustainable Agriculture, National Chung Hsing University, Taichung 40227, Taiwan

*Correspondence to:* Su-Chin Chen (scchen@nchu.edu.tw)

**Abstract.** Rapid changes in river geomorphology can occur after being disturbed by external factors like earthquakes or large dam obstructions. Studies documenting the evolution of river morphology under such conditions have advanced our understanding of fluvial geomorphology. The Dajia River in Taiwan presents a unique example of the combined effects of a coseismic fault (the 1999 Mw 7.6 Chi-Chi earthquake) and a dam. As a result of the steep terrain and abundant precipitation, rivers in Taiwan have exhibited characteristic post-disturbance evolution over 20 years. This study also considers two other comparative rivers with similar congenital conditions: the Daan River was affected by a thrust fault Chi-Chi earthquake, too; the Zhuoshui River was influenced by dam construction finished in 2001. The survey data and knickpoint migration model were used to analyze the evolution of the three rivers and propose hypothesis models. Results showed that the mobile knickpoint migrated upstream under the influence of flow, while the dam acted as a fixpoint, leading to an increased elevation gap and downstream channel incision. Thereby, the Dajia river narrowing and incision began at both ends and progressively spread to the whole reach under the combined effects.

**KEYWORDS**: dam obstruction; fixpoint; coseismic uplift; knickpoint; soft bedrock incision; river evolution



## 1. Introduction

Natural tectonic movements and artificial structures are the main factors that disturb river equilibrium. These external
influences often interact complexly; therefore, distinguishing between anthropogenic and natural drivers of landscape
evolution is difficult. In addition, changes in these external conditions, in turn drive adjustments in the riverbed, generating
new landscape patterns. River morphological development generally reflects the geology and flow stress conditions (Lyell,
1830). When a significant external impact occurs, a knickpoint (a localized discontinuity in the longitudinal profile of the
riverbed) often forms (Holland, 1976). Knickpoints can range in scale from a single waterfall to a zone of several kilometers
(Crosby and Whipple, 2006) and may result from natural factors such as extreme weather, sea-level fall, and earthquake-
induced surface rupture (Seidl and Dietrich, 1992; Whipple, 2004, Bishop et al., 2005; Heijnen et al., 2020).
The active fault causes a prominent knickpoint in stream, known as tectonic uplift, leading to a local increase in channel
steepness (Hayakawa et al., 2009; Huang et al., 2013; Cook et al., 2013). The sudden elevation change in the riverbed divides
the river profile into two reaches with differing slopes, altering the base level of fluvial erosion. The increasing flow stress
erodes the knickpoints, causing it to migrate upstream-ward over time. A long duration is required for the fluvial response to
adapt to localized surface uplift or depositional blockage by knickpoint retreat and migration upstream with time, cutting a
narrow channel and even forming a canyon. The migration process and speed are highly variable and depend on the tectonic
setting and physical nature of the riverbed (Wipple et al., 2004). The emergence and migration of knickpoints caused by
disturbance from external conditions was studied extensively (Whipple, 2001; Whipple and Trucker, 2002; Crosby and
Whipple, 2006; Clark, 2014; Ahmed et al., 2018).
Anthropogenic factors, such as reservoir construction, which is one of the most common ways humans interfere with river
hydrology and sedimentation (Magilligan and Nislow, 2005; Petts and Gurnell, 2005; Graf, 2006; Nelson et al., 2013; Liro,
2017, 2019; Zhou et al., 2018). Dam as a fixpoint in the river influences two critical components of river geomorphology: the
sediment transport capacity of the flow and the oncoming sediment load (Williams and Wolman, 1984). If the sediment
transport capacity exceeds the oncoming sediment load, the amount of sediment may be insufficient to maintain the riverbed
level, and erosion may occur. Conversely, if the sediment load exceeds the sediment transport capacity, deposition on the
riverbed would be expected to occur. The self-adjustment mechanisms of river channels responding to insufficient or excess
sediment (Brandt, 2000) results in the change in cross-section geometry, bed material size, river pattern, and slope. Previous
studies on the evolution of areas downstream of dams have primarily analyzed changes in downstream sandbars over large




spatial scales (Horn et al., 2012; Słowik et al., 2018; Kong et al., 2020) or the ecology of the lower reaches in front of dams
(Kingsford, 2000; Braatne et al., 2008; Shafroth et al., 2016). There have been few studies of exposed bedrock based on long-
term observations (Inbar, 1990). In most cases, a dam effectively traps the sediment supply from the watershed. If sediment
transfer to the downstream reaches of the dam is reduced, the armor layers of the riverbed are lost, which may cause an incision
of the fluvial channel (Surian and Rindai, 2003). This incision subsequently narrows the river cross-sections and lowers the
thalweg level.
Decades or hundreds of years are generally required for a riverbed to reach a new equilibrium after disturbance by external
conditions, so it is difficult to understand such changes based on short-period observational data (Howard et al., 1994; Tomkin
et al., 2003). Because of the abundant rainfall brought by typhoons and monsoons, the river terrain in Taiwan can alter
dramatically over a short period of time. Moreover, dams in Taiwan are built primarily in steep reaches, enhancing the rapid,
remarkable morphological evolution of the downstream reaches. The reservoirs of dams constructed on the rivers become
silted up, resulting in a lack of sediment downstream in the meantime, which causes loss of armor layers, exposure of soft rock,
and severe erosion. Another factor influencing the distinctive characteristics of Taiwanese rivers is the geological location;
Taiwan is located in a plate junction zone that experiences frequent earthquakes such as the Chi-Chi Earthquake of 1999 (Lin
et al., 2001; Ota et al., 2005), which caused the offset of Chelungpu thrust fault in central Taiwan. The surface rupture and
uplift induced the formation of knickpoints and river gorges. Twenty years later, the undercutting trend of the active channel
below dams and the migration of post-earthquake knickpoints have caused the rivers to evolve into their present forms. This
rapid evolution of river morphology over a short time makes Taiwan rivers suitable as case studies. The Dajia River is a unique
example, as a dam structure and coseismic uplift impact it simultaneously in a short reach. The current work aims to clarify
the river changes caused by the earthquake and a dam, and to propose a hypothesis for the evolution model. To compare the
various morphological developments under different external conditions, the Daan, Zhuoshui, and Dajia rivers in central
Taiwan are considered in this study.
**2.   Study area, materials, and methods**
The longitudinal changes of the river bed and the accompanying river pattern changes are the objects of observation. A
common type of longitudinal profile development for knickpoint retreat is illustrated in Fig. 1a (Gardner 1983; Whipple and
Trucker, 1999; Parker and Izumi 2000; Alonso et al. 2002; Bressan et al., 2014). As the base level of erosion fell and the abrupt
slope caused acceleration of the flow, the stream eroded the bed. During this process, apparent upstream degradation and



downstream aggradation occurred. The knickpoint migrated upward with time, companying by slope replacement. After the
river had reached a new equilibrium in a channelized pattern, the slope replacement resulted in a natural profile. During the
adjustment, the incision trend gradually slowed, and sedimentation may commence downstream (dashed line in Fig. 1a). The
profile evolved from a concave curve to a graded profile (Chamberlin and Salisbury, 1904). The well-known result of dam
construction is the progressive loss of the armor layer in the neighboring downstream river (Fig. 1b). The scouring baseline
extended downstream-ward from the dam (Olsen, 1999; Choi et al., 2005; Słowik et al., 2018). Because of the fixpoint, the
local slope at the dam toe became steeper progressively, and the dam caused the downstream river profile to be gentle and
sediment transport to decrease.
However, significant changes in the longitudinal profile must also be accompanied by variations in river patterns, which
have yet to receive much attention. Furthermore, the interaction between fault scarps and dam obstructions within a river reach
is rarely observed and studied. To address these gaps, we collected historical data for three rivers in Taiwan (Daan, Zhuoshui,
and Dajia), each representing the individual effects of faults and dams, as well as their combined effects.
**2.1    Study area**
Taiwan's climate is strongly affected by the western Pacific tropical cyclone. There are approximately three to four
typhoons and heavy rain events yearly, and the average annual precipitation is about 2500 mm. The heavy rains during the
monsoons and typhoons cause dramatic changes to riverbeds over short periods of time. In addition, because Taiwan is located
at the compressive tectonic boundary between the Eurasian and Philippine Sea plates, the collision of the two continental plates
causes tectonic breakage of the strata. On September 21, 1999, the Chi-Chi earthquake ($M_w = 7.6$) resulted in uneven uplift in
the island. Three central Taiwan rivers illustrate dams or faults' effects (Figure 2): The Daan River has been affected by vertical
fault scarps, the Dajia River by both fault scarps and a dam, and the Zhuoshui River by dam obstruction. These three important
rivers have very similar characteristics: their east-to-west flow direction; their range of elevation from sea level to ~3000 m;
their steep river slopes (the average slopes of the middle and upper reaches are greater than 1/60); and the presence of soft
rock in the mid-stream (as shown in the pink region in Figure 2). The locations of the three rivers and the Chelungpu thrust
fault are marked in Figure 2. The southern termination of the fault crosses the Zhoushui River trending north–south; the
northern termination near the Dajia and the Daan rivers shows a complex deformation pattern trending NE–SW to E–W (Lee
et al., 2002), composed of several parallel thrust faults. In the three studied reaches, the Pleistocene sedimentary rocks are
mainly composed of soft rocks consisting of sandstone, siltstone, shale, and mudstone. These rocks are generally poorly



lithified and weakened by a high water content; therefore, their resistance to water erosion is poor. The riverbed rock is readily
incised by flooding flow when the upper armoring protective layer was lost (Huang, 2014).
The Chi-Chi earthquake produced a surface rupture 80 km long (Lee et al., 2002). Several fracture planes at the north
end of the fault caused uneven uplift in the region. One of the ruptures passed through the right bank of the Shigang Dam on
the Dajia River, causing serious damage to the dam structure. The maximum vertical displacement of the surface rupture was
9 m, increasing the drop height of the bed level between the face and the back of the dam markedly. The repaired Shigang
Dam was intended to store $2.4 \times 10^6$ m$^3$ of water after the Chi-Chi earthquake; however, owing to deposition in the reservoir,
only $\sim 1.4 \times 10^6$ m$^3$ of water can now be retained. The original armor layers on the riverbed in front of the Shigang Dam were
lost rapidly, and the soft bedrock was exposed. The two rupture surfaces at the north end of the Chelungpu Fault uplifted a 1
km reach of bed in the Daan River, with a maximum vertical uplift of 10 m.
Although the southern end of the Chelungpu Fault passes downstream of the Jiji Dam (Zhuoshui River), the fault uplifted
the bed level by $\sim 2$ m, less than the uplifts in the Daan and Dajia rivers. The Jiji Dam was built in 2001 (after the 1999 Chi-
Chi earthquake), is situated on the narrowest part of the Zhuoshui River, and has a maximum designed storage capacity of 10
$\times 10^6$ m$^3$. Due to the large sediment yield in the Zhuoshui River watershed, the present-day adequate water storage capacity is
only $\sim 4 \times 10^6$ m$^3$. The Jiji Dam downstream is known for its soft bedrock canyon features, formed by dam-obstructed water
scouring.
**2.2   Materials**
Analysis of the effects of faults and dams, alteration of river patterns, changes in thalweg levels, and variations in river
cross-sections are crucial to revealing the process of river evolution. SPOT-5 and SPOT-6 satellite images and orthographic
images obtained by the Center for Space and Remote Sensing Research, National Central University (CSRSR/NCU) and Aerial
Survey Office (AFASI) of Taiwan were used to assess changes in river patterns. Multiyear cross-sectional and longitudinal
profiles were established from historical surveys by the Water Resources Agency (WRA). Additional analyses of knickpoint
retreat and variations in river elevation and width were carried out. The locations of knickpoints were determined by identifying
abrupt terrain changes and the positions of splash in the images. In order to analyze the variation of channel width ($W$), depth
($D$), and aspect ratio ($W/D$), we calculated the bank-full discharge width and depth, which represents the maximum flow that
can occur in a river before water starts overflowing and spreading out onto the floodplain. We identified the river banks and
extracted channel widths from orthographic images. The banks were defined as the boundaries between the main channel and





the adjacent floodplain.
2.3    Mathematical model

The application of the mathematical model provides an abstract description of a concrete system using physical concepts

and mathematical language. A one-dimensional Exner equation (Exner, 1925) is used to describe the advective and diffusive
knickpoint migration (Bressan et al., 2014):
$\frac{\partial z}{\partial t} + \frac{1}{(1-p_s)}\frac{\partial q_s}{\partial x} = 0$ (1a)
where $z$ is the bed elevation along the thalweg, $p_s$ is the porosity of bed sediment, $t$ is the time, $x$ is the distance, and $q_s$ is
the sediment discharge per unit width that is estimated by the product of the surface height change $\eta$, and the knickpoint
migration rate $dx/dt$ is expressed as equation 1b.
$q_s = -\eta\frac{dx}{dt}$ (1b)

The migration rate as a sediment separation per unit area homogeneously distributed over the eroding surface is expressed

as equation (1c).
$\frac{dx}{dt} = k_d[\tau(x) - \tau_C]$ (1c)
where $k_d$ is the erodibility, $\tau$ is the bed shear stress, and $\tau_C$ is the critical shear stress of the bed material. The condition of
an obvious knickpoint face, $\tau$ should be estimated using a formula that considers knickpoint as a submerged obstacle
(equation (1d)) (Engelund, 1970).
$\tau(x) = M\tau_0\left[1 + A\frac{(z-z_0)}{H_0} + B\frac{\partial z}{\partial x}\right]$ (1d)

The factors $M$, $A$, and $B$ in equation (1d) are parameters related to localized phenomena. $\tau_0$, $z_0$, and $H_0$ are the shear

stress, bed elevation and the water depth upstream of the knickpoint. The term $B\frac{\partial z}{\partial x}$ represents the change in shear stress due
to the local slope. The shear stress in the channel section upstream of the knickpoint crest ($\tau_0 = \gamma H_0 S_0$, where $\gamma$ is the specific
weight of water changes across the knickpoint due to the abrupt change in bed topography (equation (1d)). Substituting
equations (1b)–(1d) into equation (1a), equations (2a)–(2c) were obtained in below:
$\frac{\partial z}{\partial t} - C\frac{\partial z}{\partial x} - D\frac{\partial^2 z}{\partial x^2} = 0$ (2a)
$C = \left(\frac{\eta k_d \gamma}{1-p_s}\right)S_0 M A$ (2b)
$D = \left(\frac{\eta k_d \gamma}{1-p_s}\right)S_0 H_0 M B$ (2c)
where the coefficients of the first- and second-order spatial derivatives, $C$ and $D$, are known as the advection and diffusion



coefficients, respectively. It can be concluded that the key controls of the knickpoint retreat are the channel slope, the erodibility
of the bed of the river reach, the knickpoint face height, and the upstream water depth. Therefore, the present equation is a
physical-based model that can be solved with the second-order accurate implicit finite difference scheme which was
implemented in MATLAB.
**3.    RESULTS**
**3.1    Fault effect on Daan River canyon**
The scarps across the Daan River that were uplifted by the Chi-Chi earthquake caused a dramatic change in the topography,
disturbing the dynamic equilibrium of the fluvial system. Cook et al. (2013) proposed that the knickpoint propagated rapidly
after 2004 and pointed out that, after the disappearance of bedload, the tool effect caused pronounced fluvial incision of the
bedrock. Knickpoint propagation was influenced by the antiformal geological structure of the area, the presence and orientation
of interbedded strong and weak lithologies, and the proportion of discharge entering the main channel. Huang et al. (2013)
also proposed that the knickpoint retreat rate can be affected by several factors, including discharge, rock properties, geological
structures, and bedrock orientation. The channel development of the studied reach and the behavior of knickpoint retreat were
assessed by analyzing multiyear data on the form and cross-section of the river.
Successive orthographic images of the studied reach of the Daan River from 2000 to 2017 and the corresponding flow
paths are illustrated in Fig. 3. River cross-sections constructed from precise survey data are provided in Fig. 4. Chronological
longitudinal profiles of the river reach are shown in Fig. 5. Longitudinal profile data from Cook et al. (2013) were included to
make information more complete. The effect of the earthquake on the surface elevation is clearly visible in Fig. 5. In addition
to the survey data, the advective and diffusive knickpoint migration model (equation 2) was solved to mathematize the
knickpoint retreat progress after Chi-Chi earthquake. The initial condition and boundaries condition are needed to solve the
equation. The initial condition is the longitudinal profile in 1999, while the boundary conditions are the real bed changes in
upstream and downstream boundaries. The $C$ and $D$ are physical parameters and were calibrated by the survey data. In equation
2, $C$ represents the moving speed, and $D$ represents the diffusion constant. These two coefficients reflect the rate of bed erosion,
which is physically composed mainly of bed shear stress (equations 2b and 2c). Due to the actual bed erosion rates varying
with time, the parameters were adjusted to match the real changes. Before 2004, $C$ was 22.0 m/yr, and $D$ was 10.0 m²/yr; after
2004, $C$ was 91.5 m/yr, and $D$ was 18.5 m²/yr, and the simulation was continued until 2011 when the knickpoint disappear.
The result of the modeling is shown at the top left corner in Fig. 5. The knickpoint progressively retreats, companying by slope



replacement. The variation trend of the simulation and survey data is generally consistent, and the speed ($C$) has a larger value
in 2004–2011, which is also consistent with the observation.
The long-term development of the studied reach of the Daan River in the past 20 years, after the coseismic uplift, can be
divided into three periods: downstream erosion and slow knickpoint migration (earthquake to 2004); sudden migration of the
knickpoint (2004–2011); and gorge widening and eradication (2011–present).

**3.1.1 Downstream erosion and slow knickpoint migration (earthquake to 2004)**

After the Chi-Chi earthquake, coseismic ground deformation created a pop-up obstruction across the river, forming a
barrier lake behind the rupture scarp. The obstacle blocked the river flow and trapped the sediment, causing the river bed
downstream of the rupture scarp completely lose the armor layer. When the armor layer was lost, bedrock incision occurred
downstream of the uplifted zone, and the knickpoint retreat appeared. On the other hand, no significant erosion occurred
between cross-sections **a** and **b** during that period (Figs 3 and 4). A comparison of the cross-sections for 2000 and 2004 (Fig.
4) reveals that most parts of the section **a** even experienced deposition. Slight erosion in some places can be detected in the
longitudinal profiles (Fig. 5) between 1999 (after the earthquake) and 2004. Although the seismic uplift produced an obvious
knickpoint on the riverbed, that knickpoint migrated only slightly (85 m; Table 1) between 2000 and 2004. The downstream
reach of the uplifted zone showed evidence of scour, but no noticeable bedrock incision or canyon landscape had developed
yet.

**3.1.2 Sudden migration of knickpoint (2004–2011)**

The orthographic image for 2007 (Fig. 3) clearly shows that the armor layer had been removed, the bedrock had been
exposed, and the deep incision had formed a narrow channel. The knickpoint retreated upstream-ward by approximately 422
m between 2004 and 2007, accompanied by continued scouring downstream. In the uplifted reach, under the stress of the
concentrated flow in the newly formed channel, the tool effect resulted in a deepened incision of the rock bed, and a canyon
landform gradually developed. In the 2007 cross-section data for section **a**, a canyon close to the left bank can be observed,
which persisted until 2011. A rapid incision rate (5.6 m/yr) occurred in section **a**, which also experienced a narrowing rate of
about 105.5 m/yr. Bed incision and narrowing of the main channel occurred in section **b** simultaneously, with a narrowing rate
of approximately 89.9 m/yr and an incision rate of about 2.1 m/yr. Between 2007 and 2011, the knickpoint retreated upstream
by about 412 m; the incision at section **a** was lessened, but section **b** experienced a notable incision into the rock bed
accompanied by knickpoint retreat. Because an obvious gorge channel had appeared in the uplifted zone, sediment from





upstream was transported downstream, and downstream scouring transformed gradually into sedimentation; therefore, the
convex longitudinal profile was gradually erased.
**3.1.3 Gorge widening and eradication (2011 to the present)**
After 2011, the knickpoint became insignificant in the longitudinal profile, so the thalweg scouring trend slowed. The
morphology development is dominated by lateral erosion instead of vertical incision. The narrow, deep canyon evolved into a
U-shaped canyon with a wide bottom. River pattern migration from upstream caused the canyon-type channel to commence
transforming into a braided channel. The main channel of section **a** experienced deposition as a result of the sediment supply
being adequate (Fig. 5). Cook et al. (2014) proposed a mechanism of gorge eradication, called *downstream sweep erosion*,
which rapidly transformed the gorge into a beveled floodplain through the downstream propagation of a wide erosion font
located where the broad upstream channel abruptly became a narrow gorge. The sweep boundary is clearly visible in the
orthographic images for 2011 and 2017 (Fig. 3). Additional large floods are expected to cause a marked widening of the channel
instead of deepening (Huang et al., 2013). It has been estimated that removal of the gorge erosion will take 50 years (Cook et
al., 2014).
Significant incision of the channel is common after a riverbed has been uplifted suddenly by topographic tectonic
movement and the bed slope changes dramatically (Merritts et al., 1989). This was the case for the Daan River after the Chi-
Chi earthquake. After the coseismic uplift, the base level of erosion downstream reduced, so erosion increased. The river width
became notably narrower and deeper. Upward movement of the knickpoint caused the river channel in the uplifted section to
narrow rapidly. The concentrated flow caused a rapid incision of a weak geological layer in the riverbed, so the channel width
decreased sharply. Therefore, the uplifted section formed a canyon landform. As the slope at the knickpoint gradually recovered,
the incision slowed and sediment transport down the recovered river resulted in sediment deposition in the downstream channel.
The river also gradually developed lateral erosion upstream, and the river channel tended to widen. The channelization is
expected to have been swept because the sweep boundary migrated progressively downward.
**3.2    Jiji Dam effect on Zhoushui River**
Construction of the Jiji Dam on the Zhoushui River began in 1996 and operated in 2001. Orthographic images, flow paths
of the studied reach, and the locations of cross-sections **c, d**, and **e** below the Jiji Dam for 1998 to 2018 are provided in Fig. 6.
Chronological survey data of cross-sections **c, d**, and **e** are provided in Fig. 7. Chronological longitudinal profiles of the studied
reach are illustrated in Fig. 8. The river is located at the southern termination of the Chelungpu Fault (Fig. 1), where the



elevation gap caused by the earthquake is relatively small. In 1998, the Zhoushui River was a broad braided river, with many
sandbars downstream of the dam (Fig. 6). In 2003, two years after dam operation had commenced, the riverbed armor layer
had been lost and the exposed soft bedrock was clearly visible within 700 m of the toe of the dam, because of a lack of sediment.
From 2003 to 2007, the effect zone gradually expanded, and exposed bedrock extended to ~3.2 km downstream from the dam.
The bedrock's incision deepened due to the tool effect, and the flow path concentrated gradually in front of the dam. Between
2007 and 2018, the channelization and the zone with exposed bedrock expanded continuously to 6.5 km downstream of the
dam. Due to the channelization, the river cross-section became narrow and deep.

The transformation of the river and the rates of lateral and vertical change are clearly visible in the river cross-sections

(Fig. 7). There was no apparent erosion of section **c** in 2008, but the sections closer to the dam (**d** and **e**) exhibited obvious
incision (Fig. 7). After the loss of the riverbed armor layer, the flow cut down into weak bedrock. The deep main channels'
development is clearly visible in sections **d** and **e** between 1998 and 2008. During this time, the incision rate of section **e** was
around 1.2 m/yr, and the narrowing rate was around 25 m/yr. During 2008–2012, engineering measures were installed:
groundsills were added to the river channel to prevent erosion, and the riverbed level rose slightly at section **e.** However, the
channel width of section **c** was markedly narrower, with a narrowing rate of roughly 65 m/yr. Between 2008 and 2015, the
incision rates of sections **c** and **d** were roughly 1.4 m/yr. Stratified erosion is apparent in the chronological longitudinal profiles
(Fig. 8). Incision of the studied reach became increasingly severe: incision commenced at section **e** and subsequently extended
downstream to sections **d** and **c**. We infer that headward erosion did not dominate the riverbed because the Chelungpu Fault
passed through the river some distance from the dam and caused only 2 m of uplift; on the contrary, dam-induced downward
incision of the riverbed caused degradation of the reach. There is an approximately 15 m difference between the bed level of
1998 and that of 2018.

The studied reach of the Zhoushui River was a braided river prior to building of the dam. After dam construction, sediment

transport was restricted, causing loss of the armor layer downstream under the influence of the tool effect, a deeply incised
channel formed in the weak soft bedrock in front of the dam. The flow gradually became concentrated in the deep channel, the
river width decreased markedly, and the effect continued to extend downstream with time.
**3.3    The combined effect of Shigang Dam and Fault on Dajia River**

The studied reach of the Dajia River, which lies downstream of the Shigang Dam, was affected by both the dam and uplift

caused by the Chi-Chi earthquake. The Shigang Dam was broken by uneven uplift of the fault scarp across the dam (9 m on





the right side and 3 m on the left), and the downstream section **f** rose by ~7 m (see Fig. 2). The earliest knickpoint formed close
to section **f**. The base level of erosion declined downstream after uplift causing the knickpoint to move headward. During
2000–2005, the knickpoint retreated by ~400 m, and another new knickpoint formed between sections **g** and **h** (Fig. 9). The
damming effect of the Shigang Dam also caused the armor layer to be removed. The bedrock became exposed shortly after the
earthquake; however, section **f** was obviously incised during 2000–2005, whereas incision of section **g** did not occur until
2005–2008 (Fig. 10). Between 2000 and 2005, engineering measures were installed on several occasions to mitigate the
obvious erosion. Groundsills and energy-dissipation measures were constructed in front of the dam; as a result, the flow path
between section **g** and the dam became a floodplain.

The incision rate of section **g** was ~1.1 m/yr during 2005–2008, and the narrowing rate was ~47.7 m/yr. During the same

time interval, the downstream knickpoint (between sections **f** and **g**) disappeared due to river training in 2008. The knickpoint
between section **g** and section **h** retreated rapidly toward the dam (Figs 9, 11). During 2005–2008 and 2008–2017, the
knickpoint moved upstream by approximately 186 and 219 m, respectively. This retreat of the knickpoint implies that river
channel scouring did not stop. Because the riverbed strata trend northeast–southwest, flow scouring preferentially deepened
the left part of the rock bed, which moved the channel closer to the left bank. After 2008, the flow channel extended closer to
the toe of the dam. Due to the severe incision, the government started surveying section **h** after 2010 (Fig. 10). Significant
bedrock incision was recorded, with an incision rate of ~1.4 m/yr at section **h** during 2010–2017. The channel starting from
the toe of the dam was not connected with the channel caused by headward erosion from section **f** (Fig. 9) until 2017. The
2017 photograph shows a single, meandering channel that starts from the dam and runs through sections **h** and **g,** eventually
reaching section **f**, where the knickpoint had initially formed (Fig. 10). Overall, the area downstream of the Shigang Dam
displayed headward erosion of the knickpoint and incision of the rock bed in front of the dam.

In the Dajia River, the advection and diffusion equation (equation 2) was also used to represent the variation mode of

knickpoint and bed elevation. The initial condition is the longitudinal profile in 2000. The coefficients $C$ and $D$ were influenced
by bed shear stress. Due to the rapid increase in actual bed erosion rate after 2005, the parameters were adjusted to match the
actual changes. Before 2005, $C$ was 7.5 m/yr, and $D$ was 1.825 $m^2$/yr; after 2005, $C$ was 36.5 m/yr, and $D$ was 9.125 $m^2$/yr, and
the simulation was continued until 2017. The downstream boundary adopts the real bed change, while the upstream boundary
condition is fixed, considering the dam is a fixed point. The bed is progressively scoured in the nearby downstream of the dam,
and the knickpoint retreats and gradually fades away. The variation trend of the simulation and survey is generally consistent,



excluding the fact that intensive engineering works have been conducted in front of the dam to stabilize the bed.
**4.**     **Discussion**
Data on the changes in the riverbed, river width, and migration distance of the knickpoint for all three studied reaches are
provided in Table 1. Also, in Fig. 12(a), We use "**T**" symbols to represent the channel width ($W$) and depth ($D$) of the cross-
sections in three study reaches, and the aspect ratio ($W/D$) is labeled above every "**T**." After the Chi-Chi earthquake, the
channel geometry was not disturbed immediately, and the aspect ratio of the Daan River exhibited only slight changes.
Consequently, the thalweg significantly decreased with time from the downstream section; subsequently, the thalweg recovered
a little after 2011. The deepening of the upstream was slower than that downstream, but the later recovery was more obvious
in the upstream area. The aspect ratio of the Zhuoshui River dramatically declined in the upstream part after construction of
the Jiji Dam; this change extended gradually to the downstream section with time. In the Dajia River, owing to the combined
effects of the upstream dam and the earthquake, channelization of the river started at both ends of the reach and then met in
the middle. The examples of these three rivers allow us to deduce the evolution of knickpoint retreat and transformation of the
river pattern under the influence of dams and/or uplift.
The river pattern of knickpoint retreat is illustrated in Fig. 12(b), and it was also observed in the Daan River. During the
knickpoint retreat, the tool effect caused the river to narrow dramatically. However, after the river had reached a new
equilibrium in a channelized pattern, the slope replacement resulted in a natural profile. The incision trend gradually slowed
during the adjustment, and sedimentation may commence downstream (dashed line in Fig. 12(b)). The profile evolved from a
concave curve to a graded profile (Chamberlin and Salisbury, 1904). In the case of the Daan River, the topography of the
upstream gorge was gradually swept away, and the river pattern may be slowly restored to the original braided plain.
Before construction of the Jiji Dam, the studied reach of the Zhoushui River was a broad braided river. The river armor
layer was lost due to sediment trapping by the dam. Under the influence of the tool effect, the flow path in front of the dam
gradually narrowed (Fig. 12(c)). The scouring boundary extended downstream-ward from the dam. Because of the immovable
knickpoint, the local slope at the dam toe became steeper, and the dam (acting as a non-erasable knickpoint) caused the river
profile and sediment transport to remain non-equilibrium state.
The reach downstream of the Shigang Dam on the Dajia River was simultaneously affected by coseismic uplift and the
incision of a deep path in the soft rock in front of the dam. The knickpoint caused by fault uplift retreated upward with time.
Although the uplift of the Dajia River was similar to that of the Daan River, the Shigang Dam (fixpoint) restricted knickpoint



retreatment in the Dajia River, and led to scouring downward from the dam site. Therefore, we saw the river narrowing at the
two ends of the affected reach, then progressively extending to the middle, as shown in Fig. 12(d). The knickpoint caused by
the earthquake was gradually removed, but the effect of the dam remains. Therefore, the restoration of the Daan River cannot
be seen in the Dajia River.

Overall, there are apparent differences in the morphological changes to rivers caused by natural and human factors. A

knickpoint formed by fault-induced riverbed uplift is a moving point: as the knickpoint moves, the riverbed evolves gradually
from an unstable state to an equilibrium. Topographic development is like the process of childhood to old age (Davis, 1899).
In contrast, a dam can be regarded as a fixpoint on the river. The flow from the spillway outlet hits the riverbed continuously,
resulting in a decline of the erosion base level; therefore, downward erosion commences from the toe of the dam. To summarize,
changes resulting from natural tectonic movements of a riverbed may achieve equilibrium with time, whereas imbalance
caused by anthropogenic structures may be enhanced with time. Therefore, we inferred a schematic diagram of longitudinal
profile development for the combined effects as shown in Fig. 13.
**5.    Conclusions**

The Daan River, Zhoushui River, and Dajia River in central Taiwan exhibited changes in river morphology after

disturbance by earthquake uplift and dam obstruction during the past 20 years. The Daan River was affected by a thrust fault;
the Zhuoshui River was influenced by dam obstruction; and the Dajia River was both fault- and dam-influenced. In the Daan
River, the greater slope accelerated the flow velocity and drove knickpoint retreat after removal of the armor layer, resulting
in the progress of slope replacement. However, the incision faded with time, sediment deposition commenced, and the river
showed potential for recovery. Because of sediment trapping by the Jiji Dam, the Zhoushui River has transformed from braided
to gorge. The channelization started from the dam and expanded downward, and the incision progress caused the local slope
at the toe to become steeper. Because the dam acts as an immovable knickpoint, the river's sediment equilibrium could not be
re-established. The Shigang Dam on the Dajia River also caused a downward incision. The incision from the toe of the dam
subsequently connected with the knickpoint retreat caused by headward erosion from downstream, forming a single,
meandering channel at the front of the dam.

Knickpoints resulting from fault-induced riverbed uplift are moving points: as the knickpoint moves, the riverbed

evolves gradually from an unstable state to an equilibrium state. In contrast, a dam, as a fixpoint on the river, causes continuous
degradation. When both effects exist on a reach, the impact of the knickpoint gradually fades away, but the results of the dam



343 on the river persist.

**Author contribution.**

345 The following contributions were made by the authors: HEC was involved in methods development, modeling, data

346 analysis, discussion, and paper preparation. YYC participated in data analysis, discussion, and paper preparation. CYC

347 conducted the field survey, collected and analyzed data. SCC contributed to the hypothesis, concept, research design,

348 conclusions, and paper preparation.

**Competing interests.**

350 The authors declare that they have no conflict of interest.

**Acknowledgements.**

352 The Ministry of Science and Technology, Taiwan, partially supports this research under grant No. 111-2625-M-005-001.

353 The authors would like to thank AFASI, MOST, and CSRSR/NCU for supplying satellite imagery data, and thank WRA for

354 supplying river measurement data.



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





Table 1 Characteristics of the studied reaches of the Daan, Zhuoshui, and Dajia rivers

| River | Time interval | Section | Bed Change (m) | (m yr⁻¹) | Channel Widening (m) | (m yr⁻¹) | Knickpoint retreat (m) | (m yr⁻¹) | C (m yr⁻¹) |
|---|---|---|---|---|---|---|---|---|---|
| Daan | 2000–2004 | a | −0.60 | −0.15 | −103.77 | −25.94 | 85 | 21.25 | 22 |
| | | b | −1.76 | −0.44 | 47.50 | 11.88 | | | |
| | 2004–2007 | a | −16.67 | −5.56 | −316.50 | −105.50 | 422 | 140.67 | |
| | | b | −6.20 | −2.07 | −269.82 | −89.94 | | | 91.5 |
| | 2007–2011 | a | 2.06 | 0.52 | 19.30 | 4.83 | 412 | 103.00 | |
| | | b | −7.11 | −1.78 | −64.19 | −16.05 | | | |
| | 2011–2016 | a | −0.45 | −0.09 | 31.19 | 6.24 | -- | -- | -- |
| | | b | −0.84 | −0.17 | 41.27 | 8.25 | | | |
| Zhuoshui | 1998–2008 | c | −0.46 | −0.05 | −96.22 | −9.62 | | | |
| | | d | −2.24 | −0.22 | −130.41 | −13.04 | -- | -- | -- |
| | | e | −11.59 | −1.16 | −246.32 | −24.63 | | | |
| | 2008–2012 | c | −5.44 | −1.36 | −258.44 | −64.61 | | | |
| | | d | −2.77 | −0.69 | 18.43 | 4.61 | -- | -- | -- |
| | | e | 3.00 | 0.75 | 5.22 | 1.31 | | | |
| | 2012–2015 | c | −4.46 | −1.49 | −171.56 | −57.19 | | | |
| | | d | −6.65 | −2.22 | −133.24 | −44.41 | -- | -- | -- |
| | | e | −4.94 | −1.65 | −73.11 | −24.37 | | | |
| | 2015–2018 | c | −0.84 | −0.28 | 13.57 | 4.52 | | | |
| | | d | −0.86 | −0.29 | 1.31 | 0.44 | -- | -- | -- |
| | | e | −3.03 | −1.01 | 8.70 | 2.90 | | | |
| Dajia | 2000–2005 | f | −2.39 | −0.48 | −14.12 | −2.82 | 40 | 8.00 | 7.5 |
| | | g | −2.02 | −0.40 | −116.44 | −23.29 | | | |
| | 2005–2008 | f | −2.57 | −0.86 | −39.90 | −13.30 | 186 | 62.00 | |
| | | g | −7.50 | −2.50 | −142.97 | −47.66 | | | |
| | 2008–2014 | f | −1.33 | −0.22 | 12.28 | 2.05 | | | |
| | | g | −0.38 | −0.06 | 2.21 | 0.37 | | | 36.5 |
| | 2010–2014 | h | −4.20 | −1.05 | −25.45 | −6.36 | 219 | 24.33 | |
| | 2014–2017 | f | −1.39 | −0.46 | −10.44 | −3.48 | | | |
| | | g | −3.32 | −1.11 | 8.84 | 2.95 | | | |
| | | h | −5.27 | −1.76 | −20.63 | −6.88 | | | |




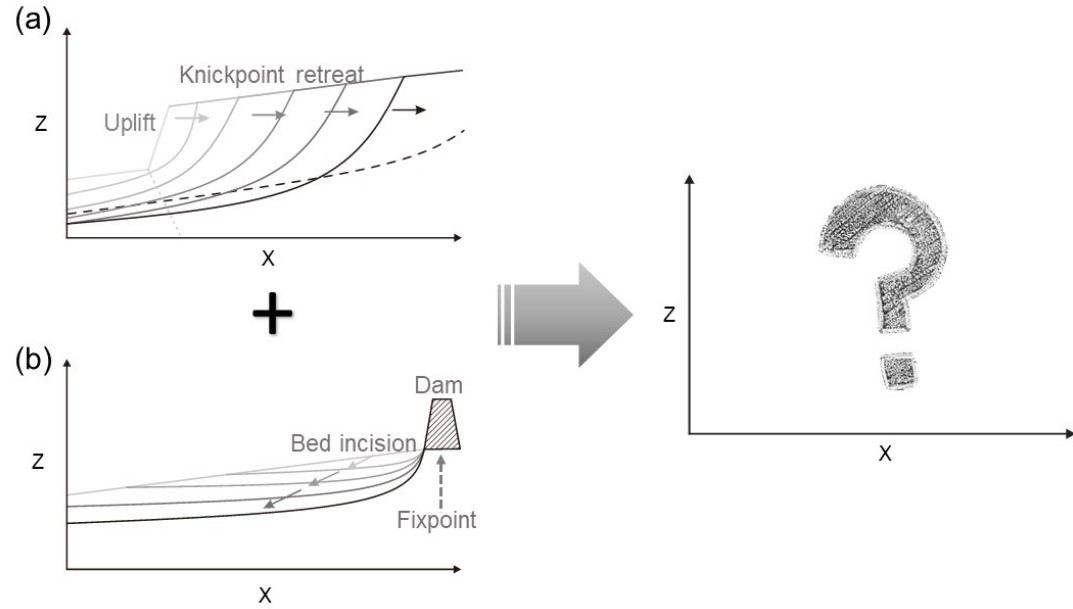


**Figure 1: Schematic diagrams of longitudinal profile development for (a) fault scarp's knickpoint, (b) dam's fixpoint,**
**and (c) How will the combined effects develop longitudinal profile?**



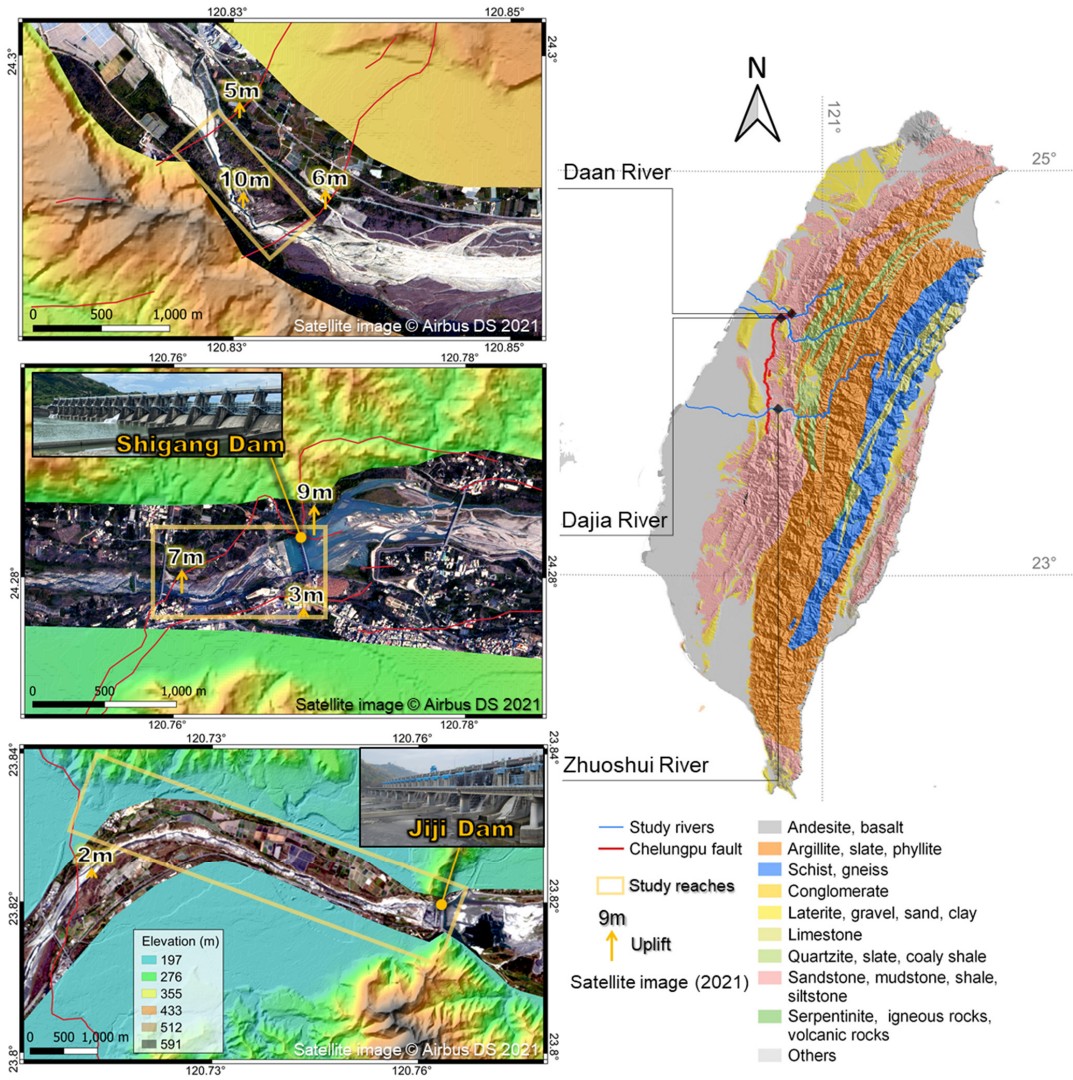

**Figure 2: Locations of the Chelungpu Fault, the three studied rivers, and satellite images (from CSRSR/NCU) showing the studied reaches.**





**Figure 3: Orthographic images (2000–2011), satellite image (2017) and flow paths of the studied reach of the Daan**
**River from 2000 to 2017.**


Earth **Surface**
**Dynamics**
Discussions
EGU

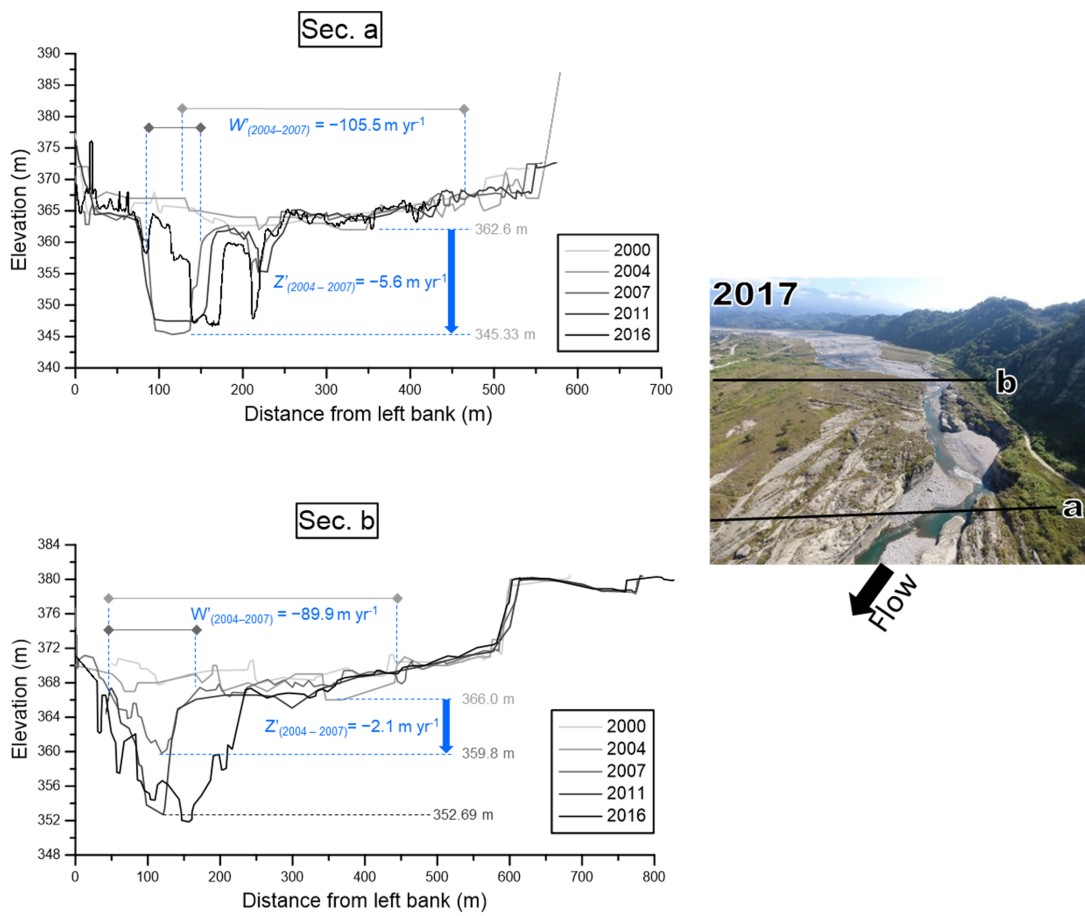


**Figure 4: Cross-sections a and b of the Daan River from 2000 to 2016 (from WRA).**




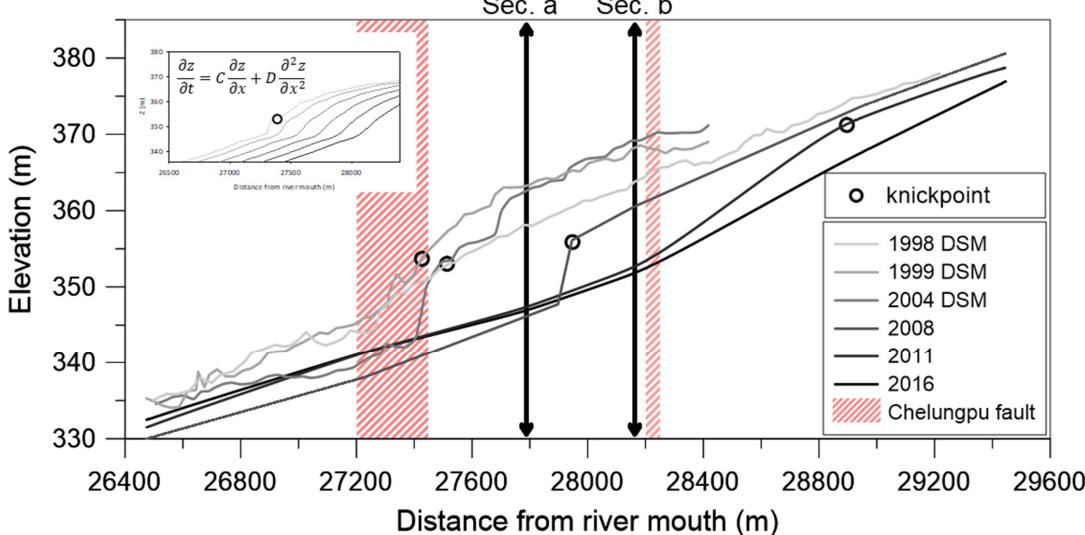


**Figure 5: Longitudinal profiles of the studied reach of the Daan River from 2000 to 2016. Profiles for 1998–2008 are from Cook et al. (2013), and 2011–2016 are from WRA. Data between 1998 and 2004 are derived from aerial photograph generated Digital Surface Models (DSMs). Knickpoint retreats are simulated using the advective-diffusive model at the top left.**



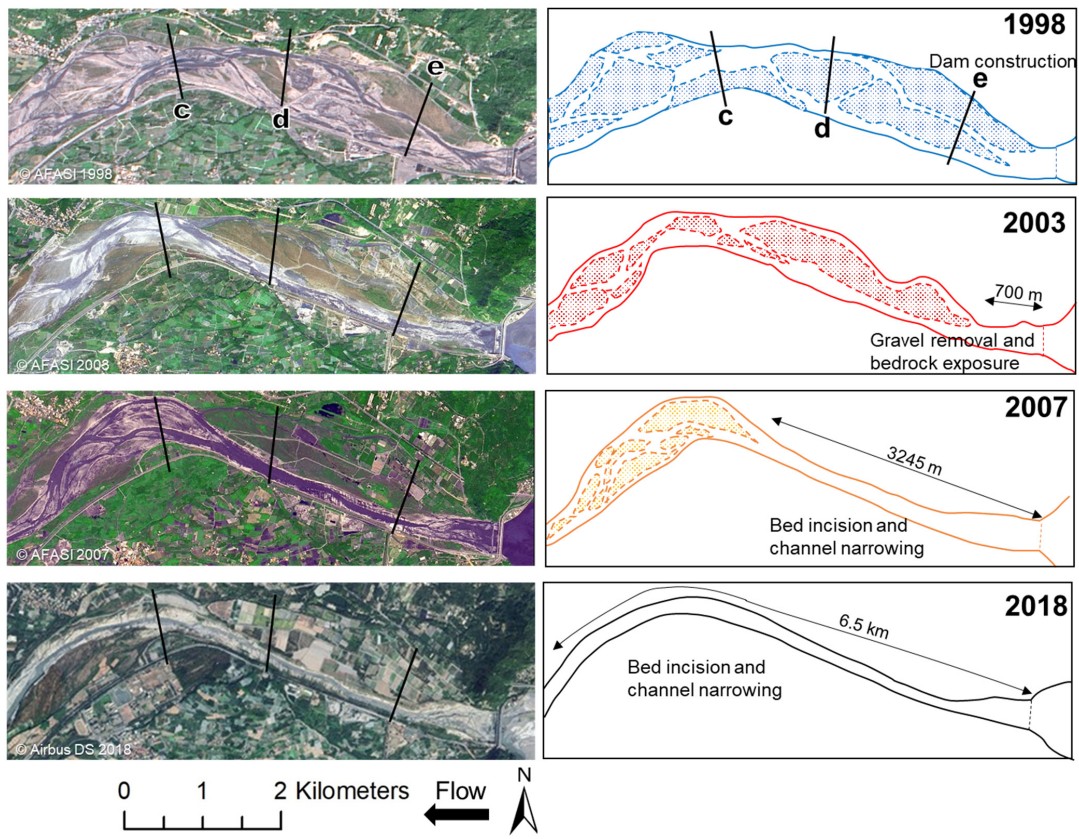

**Figure 6: Orthographic images (1998–2007), satellite image (2018), and flow paths of the studied reach of the Zhuoshui River from 1998 to 2018.**



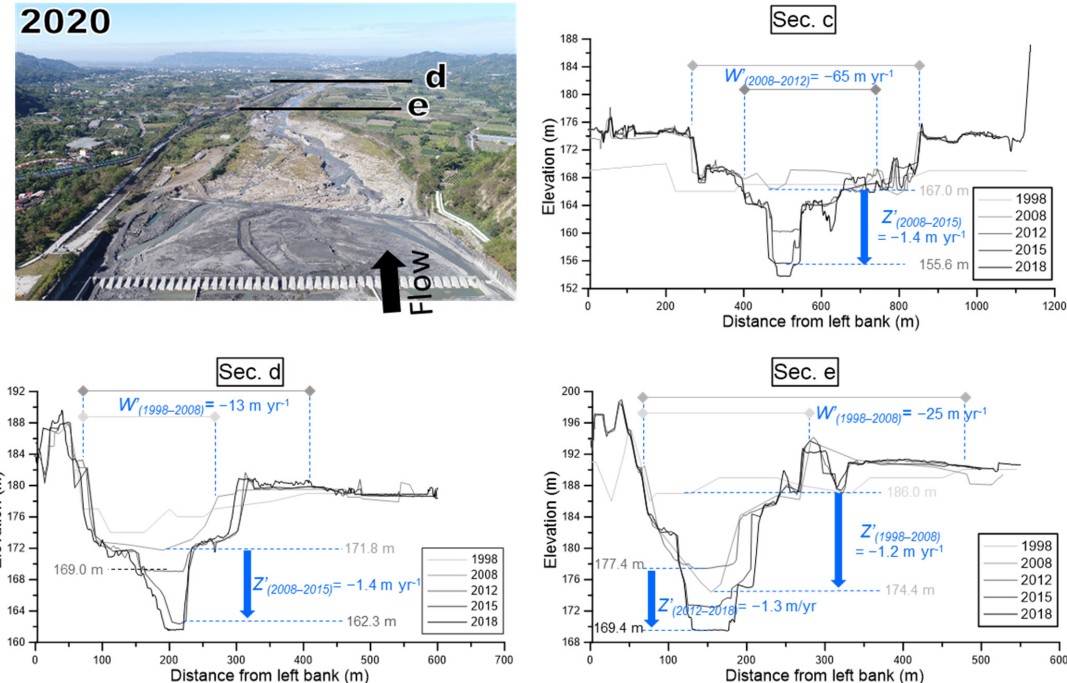

**Figure 7: Profiles of cross-sections c, d, and e of the Zhuoshui River from 1998 to 2018 (from WRA).**

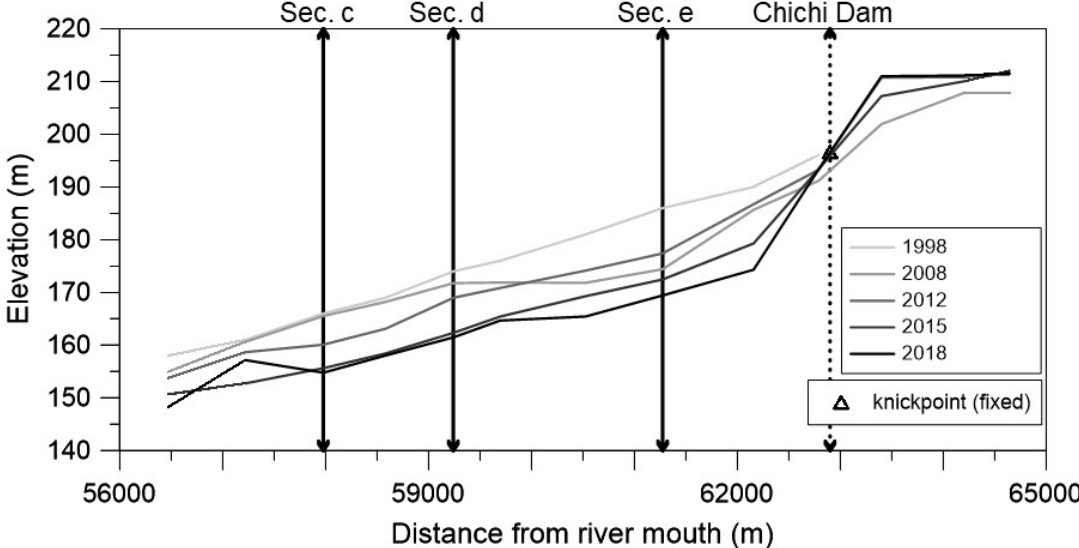

**Figure 8: Longitudinal profiles of the studied reach of the Zhuoshui River from 1998 to 2018 (from WRA).**



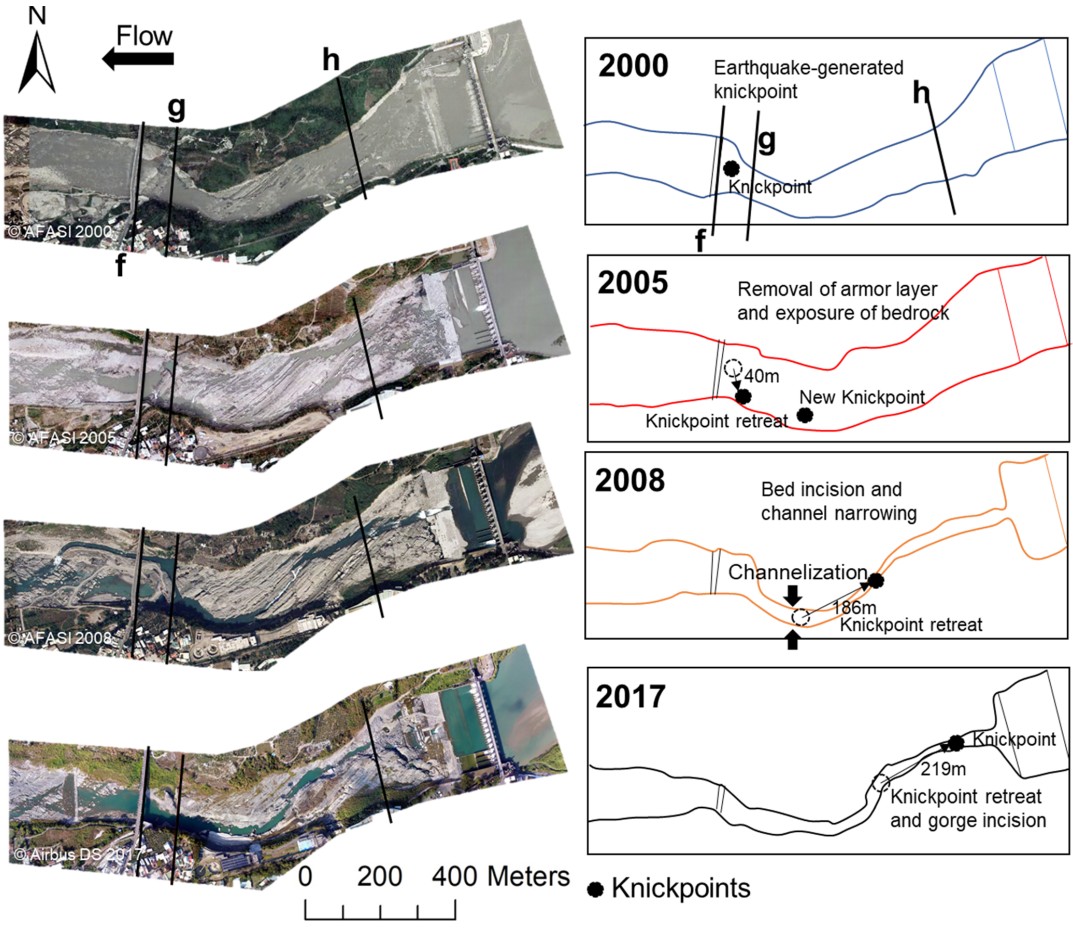

**Figure 9: Orthographic images (2000–2008), satellite image (2017), and flow paths of the studied reach of the Dajia**
**River from 2000 to 2017.**



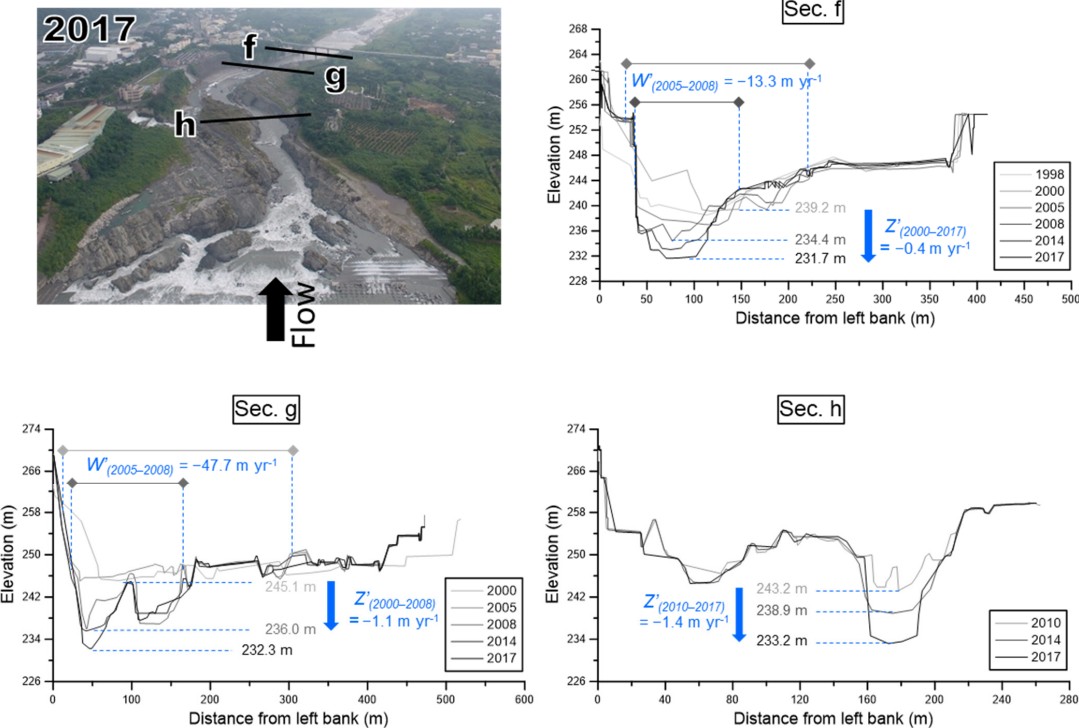

Figure 10: Cross-sections f, g, and h of the Dajia River from 2000 to 2017 (from WRA).

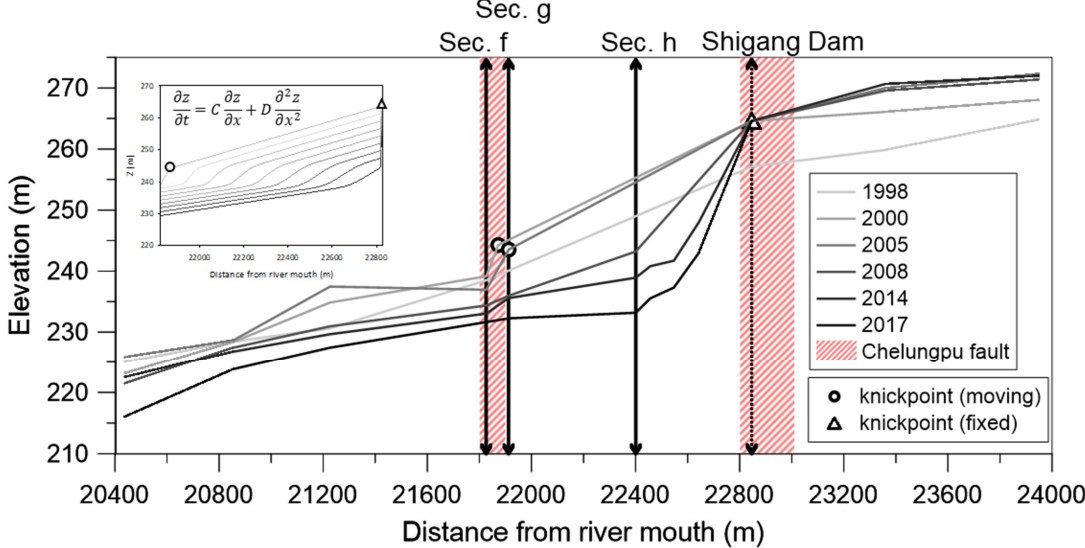

Figure 11: Longitudinal profiles of the studied reach of the Dajia River from 1998 to 2017 (from WRA). Knickpoint

retreats are simulated using the advective-diffusive model at the top left.





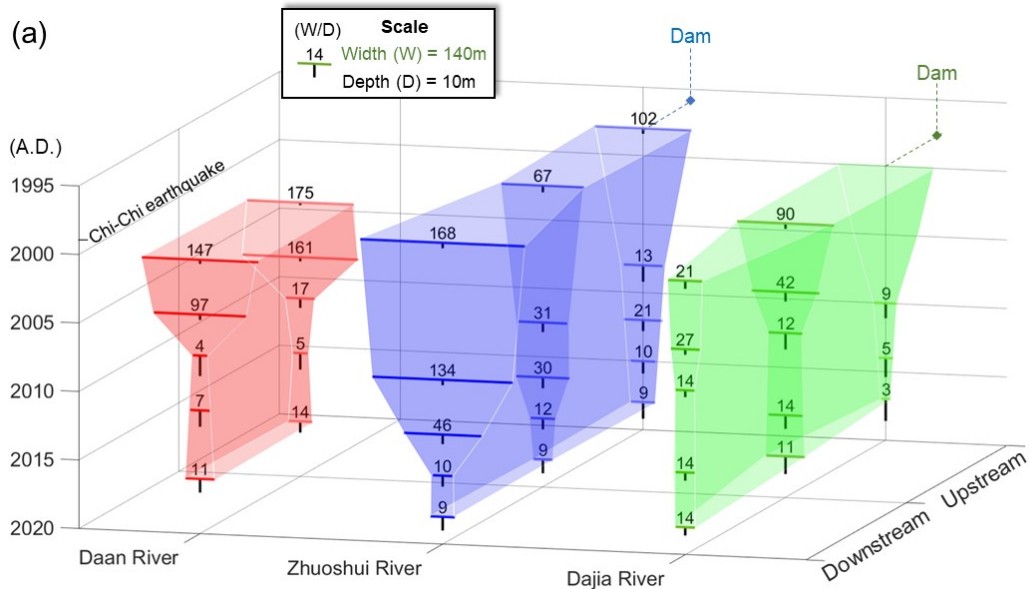


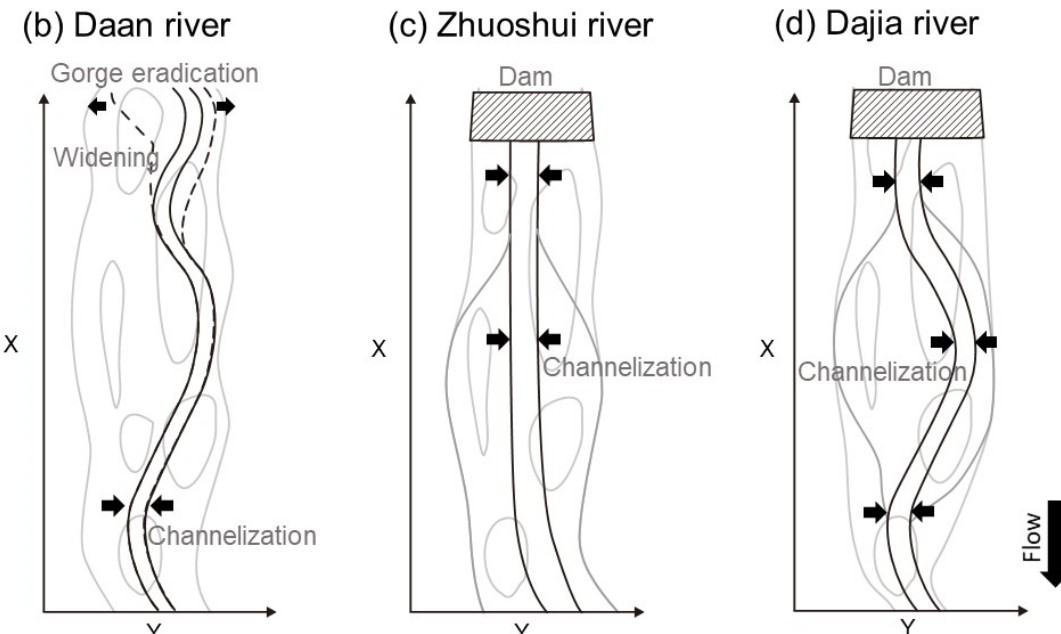


Figure 12: (a) Channel width (*W*), depth (*D*), and aspect ratio (*W/D*) of the studied reaches of the three rivers. The
aspect ratio was defined as the ratio of the bankfull width to the depth of the bankfull channel. The vertical axis shows
the time from 1995 downward to 2020, the horizontal axis shows the rivers, and the normal axis shows the sections
from downstream to upstream. Schematic diagrams of knickpoint retreat and river pattern development for (b)
coseismic uplift, (c) dam obstruction, and (d) dam obstruction and coseismic uplift.



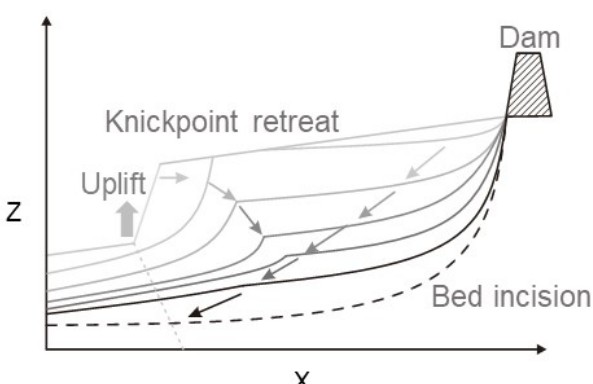


**Figure 13: A Schematic diagram of longitudinal profile development for the combined effects from dam construction**

**and coseismic uplift.**



