# Peer review of "Knickpoints and Fixpoints: The Evolution of Fluvial Morphology under the Combined Effect of Fault Uplift and Dam Obstruction on a Soft Bedrock River"

_Earth Surface Dynamics, 2023_

## Referee Comment (RC2)

Chen et al. discuss the effects of co-seismic fault displacements and a dam on the evolution of longitudinal and cross-sectional forms of three bedrock rivers in Taiwan. The multi-temporal topographic data, used in this study are unique and valuable to understand the evolution of the bedrock rivers. The content of this study matches the scope of Earth Surface Dynamics; however, I believe there are some issues to be addressed before publication. I provide the key comments below and the minor comments in the attached PDF.

Key comments:

1) Further descriptions and discussions on the combined effects of fault displacement and dam obstruction are required. Because the co-seismic knickpoint in the Dajia River disappeared due to river training in 2008, the upstream newer knickpoint and the evolution of the river following the migration of the newer knickpoint are not necessarily related to the fault activity. However, the current discussion implicitly assumes the newer knickpoint is related to fault activity. I think further explanations on how the river training in 2008 changed the channel morphology and the origin of the newer knickpoint. In addition, you have to explain Figure 13 in more detail. Although it must be the most important outcome of this study and you clearly state proposing the evolution model is the aim of this study, I do not see any descriptions on this figure.

2) It is not clear how you evaluated results of the 2-D river profile evolution. Although I am not familiar with the mathematical model in this study, the modeled river profiles do not seem to be consistent with actual river profiles. In the Daan river (fig 5), while the model predicts the river profile downstream of the knickpoint keeps its original shape during knickpoint retreat, the actual river profile changes its shape during knickpoint retreat (the knickzone stretches horizontally, and there is much less incision downstream of the knickpoint). In the Dajia River (fig 11), while the model predicts the channel slope upstream of the knickpoint is essentially the same except at the dam, the actual channel steepening occurred more extensively between the knickpoint and the dam.

3) Since the observations in the Daan river are similar to those presented in Cook et al. (2013) and Cook et al. (2014), you may want to clarify the difference between this study and previous studies or add newer implications. Although the data in 2017 is not included in Cook et al. (2014), it seems the current explanations on the Daan river are essentially the same as those in Cook et al. (2014).

Minor comments:

Line 45: What does "river pattern" mean?

Line 72-73: "abrupt slope" means "abrupt change in slope"?

Line 84: Please clarify "Historical data".

Line 95: Please indicate the location of "the middle and upper reaches".

Line 102: Huang et al. (2014)?

Line 104: Please clarify "Uneven uplift".

Line 108: Please indicate when the Shigang dam was constructed.

Line 119: Please provide the resolutions and dates of the satellite images.

Line 122: Please explain more about the survey by WRA, such as survey dates, observation spacings, instruments…etc.

Line 137: Does this equation assume sediment supply from upstream is negligible or there is no deposition?

Line 145: You may want to clarify what "localized phenomena" are.

Line 160-167: This paragraph may be better placed in the Study area section because you simply summarize previous studies rather than presenting your results.

Line 172: Please indicate parameter values used in this study. The same applies to the modeling results for the Dajia River.

Line 176: I think you should clarify what C and D represent in an actual landscape. For instance, dose C correspond with the knickpoint travel speed? If so, I suggest you calculate knickpoint travel speed using the satellite images. Since the knickpoint location and the dates the images were taken are known, you can calculate the actual knickpoint travel speed and compare them with the speed predicted by the model.

Line 216: Is "erosion font" correct?

Line 221: Consider removing "topographic."

Line 238: What is "effect zone"?

Line 240: Please clarify "channelization"

Line 247: Please explain where groundsills were added.

Line 249: Please clarify "stratified erosion"

Line 255-258: This paragraph is essentially the same as in lines 235-241.

Line 263: "The base level of erosion declined downstream after uplift causing the knickpoint to move headward." I could not understand this sentence. Please reconsider the sentence.

Line 264: Is "400m" correct? I see 40m in fig9.

Line 268-269: Does "the flow path between section g and the dam became a floodplain" mean the gorge-like channel was completely filled? Based on the satellite image, it is not clear how the artificial activity changed the channel.

Line 277-278: Although you argue "The channel starting from the toe…until 2017", the channel shown in figure 9 is continuous between the dam and the fault (section f). Please rewrite the sentence or modify the figure.

Line 288-289: Although the current gap between the modeled and actual evolution of river profiles may be considered minor, as a person who are unfamiliar with the current model prediction, I feel there is a substantial difference between the model output and your observation. Please consider explaining how you evaluated the model.

Line 293: Which timeframe does "After the Chi-Chi earthquake" mean?

Line 295 "the thalweg…": Are you referring to WD ratio?

Line 304, What does "the slope replacement resulted in a natural profile" mean?

Line 312: Is "non-equilibrium state" related to topographic steady state where rates of erosion and uplift

match? How did you judge if the river reached equilibrium or not?

Line 318-319: What does "the restoration of the Daan river" mean?

Line 322: Please clarify what you meant by "topographic development… (Davis, 1899)." Presenting an abstract concept alone does not help readers to understand your argument.

Line 334: What is "potential for recovery"?

---

## Author Comment (AC1)

**Reviewer 1**

I've reviewed the manuscript by Chen et al, examining fluvial geomorphology evolution caused by the factors like earthquake and dam construction. This manuscript presents interesting study and results, and it is well-written. I only have a few comments as follows:

Thank you for your comments. We have taken your question seriously and carefully processed it. After collecting and analyzing relevant data, we are providing the following response in the hope that it addresses your concerns.

1) What is the resolution of the DSM and the data from WRA that you use and how the resolutions might affect the interpretation of knickpoint locations and migrations?

The DSM data has a 2 to 5-meter resolution, generated using specific sets of aerial photographs. On the other hand, the data from WRA is surveyed at intervals of 5 to 10 meters.

2) Since the study is done by investigating temporal datasets, is there any significant changes in the amount of rainfall or precipitation rate over time that might have also influence/control in fluvial erosion and deposition processes. How can the variation in precipitation (year over year) impact the fluvial geomorphology evolution?

We appreciate the reviewer's interest in our research and the several analytical suggestions provided. Among them, we acknowledge that rainfall and discharge have similar implications, but discharge holds more significance for the analysis of river morphology. Therefore, we have chosen to conduct our analysis using discharge instead of rainfall, and we will provide detailed explanations in the next response.

3) Moreover, do the amount of water flowing out of the dam is consistent year over year? If not, it might have affected the fluvial geomorphology in the downstream portion, and this should also be considered.

We collected the discharge data of outflow from Shigang Dam (Dajia River) and Jiji Dam (Daan River), and the flow data of the Daan River from July 2005 to December 2019. The cumulative flow results show that the increasing trends of the discharge in the Dajia and Zhuoshui Rivers are consistent (as shown in the figure below). Both dams serve the purpose of

controlling water levels for water supply and irrigation. The direct discharge is influenced by the variations in dry and rainy seasons, resulting in intermittent changes in the discharge. In contrast, the flow in the Daan River shows continuously and stable increasing. We observed a positive correlation between the knickpoints retreat distances and the cumulative discharge in the Dajia River and also in Daan River. However, the correlation between flow and retreat distance does not exist when comparing different rivers. Additionally, A relationship between discharge and the changes in channel widening or the incision depth cannot be found.

[Figure]

4) Another factor that seems to be important is geology or lithology variation. It might be good to discuss the lithology variation along the rivers and how it can affect your interpretation in knickpoint locations and migration.

Our view is that although all three river sections have soft rock riverbeds composed of the same interbedded sandstone and shale, the knickpoint migration rates and rock distribution in these sections are not directly linked. The orientation of the rock layers may influence the position of the channel, but only locally. As seen in the example of the downstream of Shihgang Dam in the figure, where the orientation of the rock layers is visibly parallel to the local river channel in a short reach, indicating the influence of the rock layers on the river's position. Because shale has a faster erosion rate, making it easier for channels to be formed. However, in many cases, after shale erosion, the remaining isolated sandstone amplifies the impact of river flow, particle collisions, and vortexes on the isolated sandstone, causing fracturing

and accelerating the detachment of the block. Therefore, the river flow still exhibits a flow direction traverses different rock layers in most conditions.

---

## Author Comment (AC2)

**Reviewer 2**

Chen et al. discuss the effects of co-seismic fault displacements and a dam on the evolution of longitudinal and cross-sectional forms of three bedrock rivers in Taiwan. The multi-temporal topographic data, used in this study are unique and valuable to understand the evolution of the bedrock rivers. The content of this study matches the scope of Earth Surface Dynamics; however, I believe there are some major issues to be addressed before publication. I provide the key comments below and the minor comments in the attached PDF.

We would like to express our sincere gratitude to the reviewer for the positive feedback and acknowledgment. We greatly appreciate the minor comments provided by the reviewer, as they have been incredibly helpful in bridging any language gaps between our work and the readers. The open discussions resulting from these comments will undoubtedly enhance the clarity of the research content for interested readers. We have carefully addressed each comment and provided thorough responses.

Key comments:

1) Further descriptions and discussions on the combined effects of fault displacement and dam obstruction are required. Because the co-seismic knickpoint in the Dajia River disappeared due to river training in 2008, the upstream newer knickpoint and the evolution of the river following the migration of the newer knickpoint are not necessarily related to the fault activity. However, the current discussion implicitly assumes the newer knickpoint is related to fault activity. I think further explanations on how the river training in 2008 changed the channel morphology and the origin of the newer knickpoint. In addition, you have to explain Figure 13 in more detail. Although it must be the most important outcome of this study and you clearly state proposing the evolution model is the aim of this study, I do not see any descriptions on this figure.

If there were no earthquakes and the impact was solely due to the dam, the riverbed in this area would have experienced deepening similar to the result observed in the Zhuoshui River, without the occurrence of new knickpoints. Therefore, despite the presence of other factors (such as geological factors), the appearance of the new knickpoint (in 2005) is still attributed to the earthquake.

In addition, we appreciate the suggestion for Fig. 13. We have modified the content in Lines 323–327 that depicts Fig. 13.

2) It is not clear how you evaluated results of the 2-D river profile evolution. Although I am not familiar with the mathematical model in this study, the modeled river profiles do not seem to be consistent with actual river profiles. In the Daan river (fig 5), while the model predicts the river profile downstream of the knickpoint keeps its original shape during knickpoint retreat, the actual river profile changes its shape during knickpoint retreat (the knickzone stretches horizontally, and there is much less incision downstream of the knickpoint). In the Dajia River (fig 11), while the model predicts the channel slope upstream of the knickpoint is essentially the same except at the dam, the actual channel steepening occurred more extensively between the knickpoint and the dam.

We appreciate the reviewer for carefully comparing the numerical model results with the actual riverbed changes. However, it is essential to recognize that the numerical model is conceptual and involves several assumptions, such as not considering variations in the horizontal 2D plane of the terrain and assuming homogeneous parameters within the simulation area, among others. In addition, in the case of the Dajia River, due to the lacking of early cross-sectional survey data downstream of the dam, the presented cross-sections f, g, and h in this study did not reflect the steep slope in front of the dam. Also, there are also many engineering interventions here (in front of the dam) to prevent ongoing scouring. Even mobile-bed models used to simulate river sediment transport during flooding cannot accurately replicate riverbed changes. The numerical model cannot fully capture the actual scenario's detailed morphology and environmental conditions; it serves as a conceptual model based on physical mechanisms, providing trends rather than precise representations. We still use coefficient comparisons at the reach scale to demonstrate the discrepancies between the numerical model and the observed changes. Furthermore, it should be noted that the simulation results for the case under the combined effect of fault uplift and dam have not been presented or discussed in previous studies.

3) Since the observations in the Daan river are similar to those presented in Cook et al. (2013) and Cook et al. (2014), you may want to clarify the difference between this study and previous studies or add newer implications. Although the data in 2017 is not included in Cook et al. (2014),

it seems the current explanations on the Daan river are essentially the same as those in Cook et al. (2014).

    This is also the reason why we placed the case of the Daan River as the first example. Our research is not intended to directly compare with Cook et al. (2014); instead, we want to build upon previous studies' findings and discuss various scenarios. We believe that this approach holds more academic significance on morphology.

    As you correctly captured in the key comment #1, the Dajia River case is our paper's main focus. Therefore, we compared it with the Daan River and Zhuoshui River cases to highlight its importance and provide a comprehensive analysis. We believe this will be more valuable.

Minor comments:

1. Line 45: What does "river pattern" mean?

    In this context, the term "river pattern" refers to the plan view of a river reach observed from an aerial perspective. We will cite Leopold and Wolman (1957), who also used "river channel pattern" to discuss the same issue.

2. Line 72-73: "abrupt slope" means "abrupt change in slope"?

    Our intention was to convey the change from a gentle slope to a steep one. In order to prevent confusion among readers, we have made the necessary revisions to the relevant text. We sincerely appreciate your valuable feedback, as it helps us improve the quality of our work.

    The sentence has been revised as "As the base level of erosion fell, the stream encountered an abrupt shift in slope from gentle to steep, which significantly accelerated the flow, and subsequently led to erosion of the stream bed." (Lines72–74)

3. Line 84: Please clarify "Historical data".

    Incl. multiyear satellite images, orthographic images, cross-sectional and longitudinal profiles. (Lines 84–85)

4. Line 95: Please indicate the location of "the middle and upper reaches".

    Regarding the reviewer's comment on the unclear definition, we have made revisions by modifying this to approximate the average slope of the entire river (Line 96).

5. Line 102: Huang et al. (2014)?

    We have corrected this error in the revised version. (Line103).

6. Line 104: Please clarify "Uneven uplift".

    The coseismic movement resulted in vertical displacement of up to 8–10 meters at the northwestern corner of the rupture zone, where our study area is located. The distribution of uplift values is shown in Fig. 2, and further details are explained in Lines 107, 111–114. Lee et al. (2002) provide a more comprehensive description of the geometry and structure of the earthquake's surface ruptures.

7. Line 108: Please indicate when the Shigang dam was constructed.

    The Shigang dam construction was finished in 1977. After the earthquake, the dam reconstruction was completed in 2000. We have added some explanations in Lines 105–110.

8. Line 119: Please provide the resolutions and dates of the satellite images.

    The resolution of the satellite images and the orthographic images are 2m, and 25cm–50cm, respectively. We have added these in Lines 121–123.

    Dates for images of Zhuoshui River :19981111, 20031114, 20070629, 20180116

    Dates for images of Dajia River: 20000801, 20051103, 20080822, 20171214

    Dates for images of Daan River:20001112, 20040712, 20071025, 20110429, 20171214

9. Line 122: Please explain more about the survey by WRA, such as survey dates, observation spacings, instruments…etc.

    Survey is conducted using Total Station, GPS, and depth sounder. There is a survey point every 5–10 meters, and the elevation error must not exceed five centimeters. (lines125–126)

10. Line 137: Does this equation assume sediment supply from upstream is negligible or there is no deposition?

    A more accurate assumption is that $q_s$ can be regarded as the net sediment transport rate at this location, representing the difference between output and input sediment volumes within the control volume. This formula was proposed by Bressan et al. (2014).

11. Line 145: You may want to clarify what "localized phenomena" are.

    The factor $M$ represents the increase in shear stress. The term $A\frac{(z-z_0)}{H_0}$ means the change in shear stress due to the flow expansion downstream of the knickpoint. The term $B\frac{\partial z}{\partial x}$ represents the change in shear stress attributed to the local slope.

    Because the advective and diffusive knickpoint migration model has already been explained in Bressan et al. (2014), we prefer not to describe the parameters and the corresponding localized phenomena. Specifically, when calculating the model (equation 2a), including the values of M, A, and B is unnecessary.

12. Line 160-167: This paragraph may be better placed in the Study area section because you simply summarize previous studies rather than presenting your results.

We have carefully considered this opinion, but placing this paragraph in the study area section appears to provide an overly detailed introduction. However, we believe that even though it is not part of our research findings, it could serve as an opening statement for the migration results of the Daan River.

13. Line 172: Please indicate parameter values used in this study. The same applies to the modeling results for the Dajia River.

The parameters $C$ and $D$ values are provided in Lines 182-183 and 284. With these given values, it is possible to reproduce the numerical model results. Additionally, the values of parameter $C$, which denote the migration rate, are also listed in Table 1 and can be compared to the knickpoint retreat rate.

14. Line 176: I think you should clarify what C and D represent in an actual landscape. For instance, dose C correspond with the knickpoint travel speed? If so, I suggest you calculate knickpoint travel speed using the satellite images. Since the knickpoint location and the dates the images were taken are known, you can calculate the actual knickpoint travel speed and compare them with the speed predicted by the model.

From numerical perspective, $C$ and $D$ are the advection and diffusion coefficients, respectively. $C$ and $D$ contain hydraulic and sedimentological properties. Both parameters are directly related to the channel slope and erodibility, which are crucial factors controlling the retreat of knickpoints.

We are pleased that our viewpoint aligns with the reviewer. The values of $C$ are also documented in Table 1. Furthermore, we have analyzed the knickpoint retreat rate, which is also presented in Table 1. These two values are quite similar, and slight variations can be attributed to differences in data sources and calculation methods.

15. Line 216: Is "erosion font" correct?

It should be erosion front. Thank you for the correction.

16. Line 221: Consider removing "topographic.

We agree with the suggestion.

17. Line 238: What is "effect zone"?

It refers to the area under "tool effect." We have adjusted the sentence order for better understanding.

18. Line 240: Please clarify "channelization"

Channelization means the restriction of natural waterway due to external factors, limiting the flow section for water passage.

The same meaning can be found in these references: Williams & Wolman(1984), Huang et al. (2014), and Liro et al. (2017).

19. Line 247: Please explain where groundsills were added.

The reach between the sections **d** and **e**.

We have revised the sentence so the readers can know where the engineering measures were added.

"Between section **d** and **e**, groundsills, spur dikes, and tetrapod were added to the river channel to prevent erosion, and the riverbed level rose slightly at section **e**."

20. Line 249: Please clarify "stratified erosion"

We have modified the sentence as follows:

"Progressive erosion layer by layer is apparent in the chronological longitudinal profiles (Fig. 8)." (Lines 253–254)

21. Line 255-258: This paragraph is essentially the same as in lines 235-241.

We have removed the repeated parts.

22. Line 263: "The base level of erosion declined downstream after uplift causing the knickpoint to move headward." I could not understand this sentence. Please reconsider the sentence.

The sentence has been modified to "The earliest knickpoint formed close to section **f** and moving headward with time." (Line 262-263)

23. Line 264: Is "400m" correct? I see 40m in fig9.

It should be 40m. Thank you for the correction.

24. Line 268-269: Does "the flow path between section g and the dam became a floodplain" mean the gorgelike channel was completely filled? Based on the satellite image, it is not clear how the artificial activity changed the channel.

Our original intention is not to attribute the river channel changes solely to engineering influence. We will make modifications to avoid any misunderstandings.

In the cross-sectional view, it can be observed that before 2005, the thalweg in this section of the river was not well-determined, showing braided channels (characterized by exposed rock layers). The gorge-like morphology was not formed yet. (Lines 267–268)

25. Line 277-278: Although you argue "The channel starting from the toe…until 2017", the channel shown in figure 9 is continuous between the dam and the fault (section f). Please rewrite the sentence or modify the figure.

> The description here is about the existence of a knickpoint, causing a discontinuity in the riverbed slope. We will modify the sentence as follows: "In 2008, it can be observed that the knickpoint existed in the reach between sections **g** and **h;** therefore the slope of the channel is still discontinuous." (Lines 276–277)

26. Line 288-289: Although the current gap between the modeled and actual evolution of river profiles may be considered minor, as a person who are unfamiliar with the current model prediction, I feel there is a substantial difference between the model output and your observation. Please consider explaining how you evaluated the model.

> The numerical model cannot reflect the detail of the actual morphology and the environmental conditions; instead, it is a conceptual model based on physical mechanisms, capable of presenting trends only. At the reach scale, we still use coefficient comparisons to illustrate the differences between the numerical model and actual changes.

27. Line 293: Which timeframe does "After the Chi-Chi earthquake" mean?

> To be specific, it represents the time after the earthquake to the year 2021.

28. Line 295 "the thalweg…": Are you referring to WD ratio?

> Thank you for the correction. The aspect ratio (*W/D*) is corrected.

29. Line 304, What does "the slope replacement resulted in a natural profile" mean?

> This refers to the gradual recovery of the river slope to continuous. We have cited Fig 1a in this paragraph and modified the sentences related to the longitudinal profile changes in the Daan River: "During the change in river pattern, the longitudinal profile variation occurs simultaneously, similar to the one shown in Fig 1a."

30. Line 312: Is "non-equilibrium state" related to topographic steady state where rates of erosion and uplift match? How did you judge if the river reached equilibrium or not?

> Referring to the concept of sediment transport, equilibrium means

when the input and output of sediment in the control volume of a river reach a close balance.

In the case of the Zhuoshui River, it can be observed that the profiles downstream of the dam continue to erode deeper, indicating a non-equilibrium state (Fig1b).

31.Line 318-319: What does "the restoration of the Daan river" mean?

We have made a change to "the start of recovery to a braided river cannot happen in Dajia River." (Lines 317–318)

32.Line 322: Please clarify what you meant by "topographic development… (Davis, 1899)." Presenting an abstract concept alone does not help readers to understand your argument.

We accept the reviewer's suggestion and will remove this sentence to avoid confusion for the readers.

33.Line 334: What is "potential for recovery"?

As in comment 30, we will make the change to: "...the river showed potential for recovery to a braided river pattern." (Line 334)

Reference:

Davis, W. M.: Rivers and valleys of Pennsylvania, Geographical essays by William Morris Davis, 413-513, 1889.

Huang, M. W., Liao, J. J., Pan, Y. W., and Cheng, M. H.: Rapid channelization and incision into soft bedrock induced by human activity - Implications from the Bachang River in Taiwan, Engineering Geology, 177, 10-24, https://doi.org/10.1016/j.enggeo.2014.05.002, 2014.

Lee, J. C., Chu, H. T., Angelier, J., Chan, Y. C., Hu, J. C., Lu, C. Y., and Rau, R. J.: Geometry and structure of northern surface ruptures of the 1999 Mw = 7.6 Chi-Chi Taiwan earthquake: Influence from inherited fold belt structures, Journal of Structural Geology, 24, 173-192, https://doi.org/10.1016/S0191-8141(01)00056-6, 2002.

Liro, M.: Dam-induced base-level rise effects on the gravel-bed channel planform, Catena, 153, 143-156, https://doi.org/10.1016/j.catena.2017.02.005, 2017.

Kuo, C.-W., Tfwala, S., Chen, S.-C., An, H.-P., and Chu, F.-Y.: Determining transition reaches between torrents and downstream rivers using a valley morphology index in a mountainous landscape, Hydrological Processes, 35, e14393, https://doi.org/10.1002/hyp.14393, 2021.

Leopold, L. B. and Wolman, M. G.: River channel patterns: braided, meandering, and straight, US Government Printing Office, 1957.

Williams, G. P., and Wolman, M. G.: Downstream effects of dams on alluvial rivers1286, 1984.

---

## Referee Report (RR1)

I have read through the revised manuscript and found that many of my concerns are well-addressed. I leave some additional comments regarding the organization of the manuscript and the influences of discharge and lithology on the evolution of the channel morphology.

I understand that the current study is built upon Cook et al. (2014) and agree that revisiting the results of Cook et al. (2014) helps understand the evolution of the channel morphology. Yet, I think the current descriptions regarding Daan River includes both results and interpretation and suggest reorganizing the corresponding sections. Or, maybe you can introduce Cook et al. (2014) in Study area to avoid the mixing of results and discussion.

Regarding your reply to the comment from reviewer 1 (the fourth one), I have found the argument on the effects of lithology convincing and interesting. I believe you can further strengthen the current manuscript by adding the same argument about the lithology.

Line by line comments:

Line 199: I suggest citing Table 1 to show the consistency between the modeled and observed knickpoint retreat speed.

Line 335: The sentence lacks a verb and looks incomplete.

Line 339: Suggest changing continuously to continuous and increasing to increase.

Line 341-342: I could not understand what you meant by these two sentences. Since these sentences are the reply to the comment from reviewer1, I suppose you meant discharge variability does not affect the observed evolution of channel morphology. However, because discharge clearly dictates the knickpoint retreat speed when looking at the individual rivers, I wonder why the rates of knickpoint retreat are so different between the Dajia and Daan rivers. I do not think you need to find a clear answer to this question, but it is worth adding some sentences or a paragraph to the discussion.

Fig. 5: Maybe you should write that you used DSMs generated from aerial photos in the body text, not just in the figure caption.

Fig.12: What do background thin-colored lines represent?

Fig. 13: Is the Y-axis label "Accumulated flow"? Also, since there is no knickpoint in Zhuoshui river, I wonder why Zhuoshui river is included.

---

## Author Response (AR2)

Dear Chen Su-Chin and colleagues,

First, please accept my apology for the long delay of this editorial decision. Due to no fault of his own, our designated AE, Greg Hancock, has not been able to pursue his tasks. I have taken the unusual step of changing my role from senior editor to associate editor, and shall oversee further progress of your manuscript. I am aware of the fact that your study deals with examples on which I have worked earlier, and shall try to avoid any biass.

Greg Hancock's earlier efforts yielded two reviews of your work. Both referees are convinced that your manuscript can make a valuable contribution to the literature on erosional river dynamics. However, they highlight several issues that will require attention prior to any decision about publication.

We would like to thank the two editors for handling this manuscript, and thank you for affirming the value of the study. We have tried our best to give a detailed reply to reviewers.

Referee #1 is concerned about the influence of rainfall variations and changes in discharge over time. They also request attention to the role of lithology. It is true that the lithology of your two examples is particular, and I agree that it would be helpful to have some discussion of its effects on the portability of your results and findings.

Since rainfall and discharge are typically highly correlated, we only collected flow data to investigate the time-dependent effects. In rivers with dams, floods play a crucial role in erosion and deposition; therefore, discharge usually have a greater dominance than rainfall. The relationship between discharge and the distance of knickpoint retreat is discussed in Figure 13 and Lines 324–331. Our data supports a positive correlation between discharge and the rate of knickpoint change in single river. However, the correlation between flow and retreat distance does not exist when comparing different rivers. Additionally, a relationship between discharge and the changes in channel widening or the incision depth cannot be found.

In Lines 100–104, we have emphasized Taiwan's alluvial rivers are often characterized by the presence and exposure of soft bedrock, predominantly comprising mudstone and sandstone, with unconfined compressive strengths ranging from 0.5 to 25.0 MPa (Lai et al., 2011). Each summer, the region is

susceptible to typhoons that result in substantial river flooding. In addition, the three study reaches are all located in the soft rock region (Fig 2)

Our view is that although all three river sections have soft rock riverbeds composed of the same interbedded sandstone and shale, the knickpoint migration rates and rock distribution in these sections are not directly linked. The orientation of the rock layers may influence the position of the channel, but only locally. As seen in the example of the downstream of Shihgang Dam in the figure, where the orientation of the rock layers is visibly parallel to the local river channel in a short reach, indicating the influence of the rock layers on the river's position. Because shale has a faster erosion rate, making it easier for channels to be formed. However, in many cases, after shale erosion, the remaining isolated sandstone amplifies the impact of river flow, particle collisions, and vortexes on the isolated sandstone, causing fracturing and accelerating the detachment of the block. Therefore, the river flow still exhibits a flow direction traverses different rock layers in most conditions.

Our study is based on soft rock riverbeds, and this characteristic has allowed us to acquire a significant amount of meaningful data over 20 years. When it comes to the influence of dams and coseismic uplift on river channels, the changing trends are likely to be similar.

Referee #2 has raised some important points about the development of the Daan and Dajia knick zones. They draw attention to the manipulation of the Dajia river, and to discrepancies between the observed and modelled evolution of the Daan long profile. In addition, they ask about insights that reach beyond those that were published earlier (this is where I must refrain from comment, and simply act as go between).

Regarding the questions of the Dajia River raised by Referee #2, we have provided clarification in our response, and we have revised the manuscript (lines 269–270) to address their concerns. Additionally, the interpretation of the numerical model results has been included in lines 162–166. As for the differences from Cook et al. (2013) and Cook et al. (2014), we have explained this in our response to Referee #2 as well.

I encourage you to address these points and any others raised by the referees in their reports, and look forward to receiving a revised manuscript. It is likely that this new version will be sent out to reviewers, but we shall endeavour to keep the process lean on our side.

Best wishes,
Niels Hovius

I've reviewed the manuscript by Chen et al, examining fluvial geomorphology evolution caused by the factors like earthquake and dam construction. This manuscript presents interesting study and results, and it is well-written. I only have a few comments as follows:

Thank you for your comments. We have taken your question seriously and carefully processed it. After collecting and analyzing relevant data, we are providing the following response in the hope that it addresses your concerns.

1) What is the resolution of the DSM and the data from WRA that you use and how the resolutions might affect the interpretation of knickpoint locations and migrations?

The DSM data has a 2 to 5-meter resolution, generated using specific sets of aerial photographs. On the other hand, the data from WRA is surveyed at intervals of 5 to 10 meters.

2) Since the study is done by investigating temporal datasets, is there any significant changes in the amount of rainfall or precipitation rate over time that might have also influence/control in fluvial erosion and deposition processes. How can the variation in precipitation (year over year) impact the fluvial geomorphology evolution?

We appreciate the reviewer's interest in our research and the several analytical suggestions provided. Among them, we acknowledge that rainfall and discharge have similar implications, but discharge holds more significance for the analysis of river morphology. Therefore, we have chosen to conduct our analysis using discharge instead of rainfall, and we will provide detailed explanations in the next response.

3) Moreover, do the amount of water flowing out of the dam is consistent year over year? If not, it might have affected the fluvial geomorphology in the downstream portion, and this should also be considered.

We collected the discharge data of outflow from Shigang Dam (Dajia River) and Jiji Dam (Zhuoshui River), and the flow data of the Daan River from July 2005 to December 2019. The cumulative flow results show that the increasing trends of the discharge in the Dajia and Zhuoshui Rivers are consistent (as shown in the figure below). Both dams serve the purpose of

controlling water levels for water supply and irrigation. The direct discharge is influenced by the variations in dry and rainy seasons, resulting in intermittent changes in the discharge. In contrast, the flow in the Daan River shows continuously and stable increasing. We observed a positive correlation between the knickpoints retreat distances and the cumulative discharge in the Dajia River and also in Daan River. However, the correlation between flow and retreat distance does not exist when comparing different rivers. Additionally, A relationship between discharge and the changes in channel widening or the incision depth cannot be found.

[Figure]

4) Another factor that seems to be important is geology or lithology variation. It might be good to discuss the lithology variation along the rivers and how it can affect your interpretation in knickpoint locations and migration.

Our view is that although all three river sections have soft rock riverbeds composed of the same interbedded sandstone and shale, the knickpoint migration rates and rock distribution in these sections are not directly linked. The orientation of the rock layers may influence the position of the channel, but only locally. As seen in the example of the downstream of Shihgang Dam in the figure, where the orientation of the rock layers is visibly parallel to the local river channel in a short reach, indicating the influence of the rock layers on the river's position. Because shale has a faster erosion rate, making it easier for channels to be formed. However, in many cases, after shale erosion, the remaining isolated sandstone amplifies the impact of river flow, particle collisions, and vortexes on the isolated sandstone, causing fracturing and accelerating the detachment of the block. Therefore, the river flow still exhibits a flow direction traverses different rock layers in most conditions.

[Figure]

**Response to Reviewer 2**

Chen et al. discuss the effects of co-seismic fault displacements and a dam on the evolution of longitudinal and cross-sectional forms of three bedrock rivers in Taiwan. The multi-temporal topographic data, used in this study are unique and valuable to understand the evolution of the bedrock rivers. The content of this study matches the scope of Earth Surface Dynamics; however, I believe there are some major issues to be addressed before publication. I provide the key comments below and the minor comments in the attached PDF.

> We would like to express our sincere gratitude to the reviewer for the positive feedback and acknowledgment. We greatly appreciate the minor comments provided by the reviewer, as they have been incredibly helpful in bridging any language gaps between our work and the readers. The open discussions resulting from these comments will undoubtedly enhance the clarity of the research content for interested readers. We have carefully addressed each comment and provided thorough responses.

Key comments:

1) Further descriptions and discussions on the combined effects of fault displacement and dam obstruction are required. Because the co-seismic knickpoint in the Dajia River disappeared due to river training in 2008, the upstream newer knickpoint and the evolution of the river following the migration of the newer knickpoint are not necessarily related to the fault activity. However, the current discussion implicitly assumes the newer knickpoint is related to fault activity. I think further explanations on how the river training in 2008 changed the channel morphology and the origin of the newer knickpoint. In addition, you have to explain Figure 13 in more detail. Although it must be the most important outcome of this study and you clearly state proposing the evolution model is the aim of this study, I do not see any descriptions on this figure.

> In 2008, the river training efforts were focused on the reach between the bridge (section f) and the original knickpoint. However, the new knickpoint had already formed before 2005, and its emergence should be attributed to the co-effect of changes in river pattern and alternating erosion of rock layers (we have added the reason in lines 269–270). This led to the appearance of the new knickpoint during the process of river bed deepening. If there were no earthquakes and the impact was solely due to the dam, the riverbed in

this area would have experienced deepening similar to the result observed in the Zhuoshui River, without the occurrence of new knickpoints. Therefore, despite the presence of other factors (such as geological factors), the appearance of the new knickpoint (before 2005) is still attributed to the earthquake.

In addition, we appreciate the suggestion for Fig. 14. We have modified the content in Lines 324–328 that depicts Fig. 14.

2) It is not clear how you evaluated results of the 2-D river profile evolution. Although I am not familiar with the mathematical model in this study, the modeled river profiles do not seem to be consistent with actual river profiles. In the Daan river (fig 5), while the model predicts the river profile downstream of the knickpoint keeps its original shape during knickpoint retreat, the actual river profile changes its shape during knickpoint retreat (the knickzone stretches horizontally, and there is much less incision downstream of the knickpoint). In the Dajia River (fig 11), while the model predicts the channel slope upstream of the knickpoint is essentially the same except at the dam, the actual channel steepening occurred more extensively between the knickpoint and the dam.

We appreciate the reviewer for carefully comparing the numerical model results with the actual riverbed changes. However, it is essential to recognize that the numerical model is conceptual and involves several assumptions, such as not considering variations in the horizontal 2D plane of the terrain and assuming homogeneous parameters within the simulation area, among others. In addition, in the case of the Dajia River, due to the lacking of early cross-sectional survey data downstream of the dam, the presented cross-sections f, g, and h in this study did not reflect the steep slope in front of the dam. Also, there are also many engineering interventions here (in front of the dam) to prevent ongoing scouring. Even mobile-bed models used to simulate river sediment transport during flooding cannot accurately replicate riverbed changes. The numerical model cannot fully capture the actual scenario's detailed morphology and environmental conditions; it serves as a conceptual model based on physical mechanisms, providing trends rather than precise representations. We still use coefficient comparisons at the reach scale to demonstrate the discrepancies between the numerical model and the observed changes. Furthermore, it should be noted that the simulation results for the case under the combined effect of fault uplift and dam have

not been presented or discussed in previous studies. We have added the meaning of the numerical model in lines 162–166.

3) Since the observations in the Daan river are similar to those presented in Cook et al. (2013) and Cook et al. (2014), you may want to clarify the difference between this study and previous studies or add newer implications. Although the data in 2017 is not included in Cook et al. (2014), it seems the current explanations on the Daan river are essentially the same as those in Cook et al. (2014).

This is also the reason why we placed the case of the Daan River as the first example. Our research is not intended to directly compare with Cook et al. (2014); instead, we want to build upon previous studies' findings and discuss various scenarios. We believe that this approach holds more academic significance on morphology.

As you correctly captured in the key comment #1, the Dajia River case is our paper's main focus. Therefore, we compared it with the Daan River and Zhuoshui River cases to highlight its importance and provide a comprehensive analysis. We believe this will be more valuable.

Minor comments:

1. Line 45: What does "river pattern" mean?

In this context, the term "river pattern" refers to the plan view of a river reach observed from an aerial perspective. We will cite Leopold and Wolman (1957), who also used "river channel pattern" to discuss the same issue.

2. Line 72-73: "abrupt slope" means "abrupt change in slope"?

Our intention was to convey the change from a gentle slope to a steep one. In order to prevent confusion among readers, we have made the necessary revisions to the relevant text. We sincerely appreciate your valuable feedback, as it helps us improve the quality of our work.

The sentence has been revised as "As the base level of erosion fell, the stream encountered an abrupt shift in slope from gentle to steep, which significantly accelerated the flow, and subsequently led to erosion of the stream bed." (Lines72–74)

3. Line 84: Please clarify "Historical data".

Incl. multiyear satellite images, orthographic images, cross-sectional

and longitudinal profiles. (Lines 84–85)

4. Line 95: Please indicate the location of "the middle and upper reaches".

Regarding the reviewer's comment on the unclear definition, we have made revisions by modifying this to approximate the average slope of the entire river (Line 96).

5. Line 102: Huang et al. (2014)?

We have corrected this error in the revised version. (Line104).

6. Line 104: Please clarify "Uneven uplift".

The coseismic movement resulted in vertical displacement of up to 8–10 meters at the northwestern corner of the rupture zone, where our study area is located. The distribution of uplift values is shown in Fig. 2, and further details are explained in Lines 108, 112–115. Lee et al. (2002) provides a more comprehensive description of the geometry and structure of the earthquake's surface ruptures.

7. Line 108: Please indicate when the Shigang dam was constructed.

The Shigang dam construction was finished in 1977. After the earthquake, the dam reconstruction was completed in 2000. We have added some explanations in Lines 106–111.

8. Line 119: Please provide the resolutions and dates of the satellite images.

The resolution of the satellite images and the orthographic images are 2m, and 25cm–50cm, respectively. We have added these in Lines 122–124.

Dates for images of Zhuoshui River :19981111, 20031114, 20070629, 20180116

Dates for images of Dajia River: 20000801, 20051103, 20080822, 20171214

Dates for images of Daan River:20001112, 20040712, 20071025, 20110429, 20171214

9. Line 122: Please explain more about the survey by WRA, such as survey dates, observation spacings, instruments…etc.

Survey is conducted using Total Station, GPS, and depth sounder. There is a survey point every 5–10 meters, and the elevation error must not exceed five centimeters. (lines126–127)

10. Line 137: Does this equation assume sediment supply from upstream is negligible or there is no deposition?

A more accurate assumption is that $q_s$ can be regarded as the net sediment transport rate at this location, representing the difference between output and input sediment volumes within the control volume. This formula was proposed by Bressan et al. (2014).

11. Line 145: You may want to clarify what "localized phenomena" are.

The factor $M$ represents the increase in shear stress. The term $A\frac{(z-z_0)}{H_0}$ means the change in shear stress due to the flow expansion downstream of the knickpoint. The term $B\frac{\partial z}{\partial x}$ represents the change in shear stress attributed to the local slope.

Because the advective and diffusive knickpoint migration model has already been explained in Bressan et al. (2014), we prefer not to describe the parameters and the corresponding localized phenomena. Specifically, when calculating the model (equation 2a), including the values of M, A, and B is unnecessary.

12. Line 160-167: This paragraph may be better placed in the Study area section because you simply summarize previous studies rather than presenting your results.

We have carefully considered this opinion, but placing this paragraph in the study area section appears to provide an overly detailed introduction. However, we believe that even though it is not part of our research findings, it could serve as an opening statement for the migration results of the Daan River.

13. Line 172: Please indicate parameter values used in this study. The same applies to the modeling results for the Dajia River.

The parameters $C$ and $D$ values are provided in Lines 182-183 and 284. With these given values, it is possible to reproduce the numerical model results. Additionally, the values of parameter $C$, which denote the migration rate, are also listed in Table 1 and can be compared to the knickpoint retreat rate.

14. Line 176: I think you should clarify what C and D represent in an actual landscape. For instance, dose C correspond with the knickpoint travel speed? If so, I suggest you calculate knickpoint travel speed using the satellite images. Since the knickpoint location and the dates the images were taken are known, you can calculate the actual knickpoint travel speed

and compare them with the speed predicted by the model.

From numerical perspective, $C$ and $D$ are the advection and diffusion coefficients, respectively. $C$ and $D$ contain hydraulic and sedimentological properties. Both parameters are directly related to the channel slope and erodibility, which are crucial factors controlling the retreat of knickpoints.

We are pleased that our viewpoint aligns with the reviewer. The values of $C$ are also documented in Table 1. Furthermore, we have analyzed the knickpoint retreat rate, which is also presented in Table 1. These two values are quite similar, and slight variations can be attributed to differences in data sources and calculation methods.

15. Line 216: Is "erosion font" correct?

It should be erosion front. Thank you for the correction.

16. Line 221: Consider removing "topographic.

We agree with the suggestion.

17. Line 238: What is "effect zone"?

It refers to the area under "tool effect." We have adjusted the sentence order for better understanding.

18. Line 240: Please clarify "channelization"

Channelization means the restriction of natural waterway due to external factors, limiting the flow section for water passage.

The same meaning can be found in these references: Williams & Wolman(1984), Huang et al. (2014), and Liro et al. (2017).

19. Line 247: Please explain where groundsills were added.

The reach between the sections **d** and **e**.

We have revised the sentence so the readers can know where the engineering measures were added.

"Between section **d** and **e**, groundsills, spur dikes, and tetrapod were added to the river channel to prevent erosion, and the riverbed level rose slightly at section **e**."

20. Line 249: Please clarify "stratified erosion"

We have modified the sentence as follows:

"Progressive erosion layer by layer is apparent in the chronological longitudinal profiles (Fig. 8)." (Lines 258–259)

21. Line 255-258: This paragraph is essentially the same as in lines 235-241.

We have removed the repeated parts.

22. Line 263: "The base level of erosion declined downstream after uplift causing the knickpoint to move headward." I could not understand this sentence. Please reconsider the sentence.

   The sentence has been modified to "The earliest knickpoint formed close to section **f** and moving headward with time." (Line 267-268)

23. Line 264: Is "400m" correct? I see 40m in fig9.

   It should be 40m. Thank you for the correction.

24. Line 268-269: Does "the flow path between section g and the dam became a floodplain" mean the gorgelike channel was completely filled? Based on the satellite image, it is not clear how the artificial activity changed the channel.

   Our original intention is not to attribute the river channel changes solely to engineering influence. We will make modifications to avoid any misunderstandings.

   In the cross-sectional view, it can be observed that before 2005, the thalweg in this section of the river was not well-determined, showing braided channels (characterized by exposed rock layers). The gorge-like morphology was not formed yet. (Lines 272–273)

25. Line 277-278: Although you argue "The channel starting from the toe…until 2017", the channel shown in figure 9 is continuous between the dam and the fault (section f). Please rewrite the sentence or modify the figure.

   The description here is about the existence of a knickpoint, causing a discontinuity in the riverbed slope. We will modify the sentence as follows: "In 2008, it can be observed that the knickpoint existed in the reach between sections **g** and **h;** therefore the slope of the channel is still discontinuous." (Lines 281–282)

26. Line 288-289: Although the current gap between the modeled and actual evolution of river profiles may be considered minor, as a person who are unfamiliar with the current model prediction, I feel there is a substantial difference between the model output and your observation. Please consider explaining how you evaluated the model.

   The numerical model cannot reflect the detail of the actual morphology and the environmental conditions; instead, it is a conceptual model based on physical mechanisms, capable of presenting trends only. At the reach scale, we still use coefficient comparisons to illustrate the differences

between the numerical model and actual changes.

27. Line 293: Which timeframe does "After the Chi-Chi earthquake" mean?

To be specific, it represents the time after the earthquake to the year 2021.

28. Line 295 "the thalweg…": Are you referring to WD ratio?

Thank you for the correction. The aspect ratio ($W/D$) is corrected.

29. Line 304, What does "the slope replacement resulted in a natural profile" mean?

This refers to the gradual recovery of the river slope to continuous. We have cited Fig 1a in this paragraph and modified the sentences related to the longitudinal profile changes in the Daan River: "During the change in river pattern, the longitudinal profile variation occurs simultaneously, similar to the one shown in Fig 1a."

30. Line 312: Is "non-equilibrium state" related to topographic steady state where rates of erosion and uplift match? How did you judge if the river reached equilibrium or not?

Referring to the concept of sediment transport, equilibrium means when the input and output of sediment in the control volume of a river reach a close balance.

In the case of the Zhuoshui River, it can be observed that the profiles downstream of the dam continue to erode deeper, indicating a non-equilibrium state (Fig1b).

31. Line 318-319: What does "the restoration of the Daan river" mean?

We have made a change to "the start of recovery to a braided river cannot happen in Dajia River." (Lines 322–323)

32. Line 322: Please clarify what you meant by "topographic development… (Davis, 1899)." Presenting an abstract concept alone does not help readers to understand your argument.

We accept the reviewer's suggestion and will remove this sentence to avoid confusion for the readers.

33. Line 334: What is "potential for recovery"?

As in comment 30, we will make the change to: "... the start of recovery to a braided river cannot happen in the Dajia River." (Line 332-333)

Reference:

Cook, K. L., Turowski, J. M., and Hovius, N.: A demonstration of the importance of bedload transport for fluvial bedrock erosion and knickpoint propagation, Earth Surface Processes and Landforms, 38, 683-695, https://doi.org/10.1002/esp.3313, 2013.

Cook, K. L., Turowski, J. M., and Hovius, N.: River gorge eradication by downstream sweep erosion, Nature Geoscience, 7, 682-686, https://doi.org/10.1038/ngeo2224, 2014.

Davis, W. M.: Rivers and valleys of Pennsylvania, Geographical essays by William Morris Davis, 413-513, 1889.

Huang, M. W., Liao, J. J., Pan, Y. W., and Cheng, M. H.: Rapid channelization and incision into soft bedrock induced by human activity - Implications from the Bachang River in Taiwan, Engineering Geology, 177, 10-24, https://doi.org/10.1016/j.enggeo.2014.05.002, 2014.

Lai, Y. G., Greimann, B. P., and Wu, K.: Soft Bedrock Erosion Modeling with a Two-Dimensional Depth-Averaged Model, J. Hydraul. Eng., 137, 804–814, https://doi.org/10.1061/(asce)hy.1943-7900.0000363, 2011.

Lee, J. C., Chu, H. T., Angelier, J., Chan, Y. C., Hu, J. C., Lu, C. Y., and Rau, R. J.: Geometry and structure of northern surface ruptures of the 1999 Mw = 7.6 Chi-Chi Taiwan earthquake: Influence from inherited fold belt structures, Journal of Structural Geology, 24, 173-192, https://doi.org/10.1016/S0191-8141(01)00056-6, 2002.

Leopold, L. B. and Wolman, M. G.: River channel patterns: braided, meandering, and straight, US Government Printing Office, 1957.

Liro, M.: Dam-induced base-level rise effects on the gravel-bed channel planform, Catena, 153, 143-156, https://doi.org/10.1016/j.catena.2017.02.005, 2017.

Kuo, C.-W., Tfwala, S., Chen, S.-C., An, H.-P., and Chu, F.-Y.: Determining transition reaches between torrents and downstream rivers using a valley morphology index in a mountainous landscape, Hydrological Processes, 35, e14393, https://doi.org/10.1002/hyp.14393, 2021.

Williams, G. P., and Wolman, M. G.: Downstream effects of dams on alluvial rivers1286, 1984.

---

## Author Response (AR3)

Public justification (visible to the public if the article is accepted and published):
Firstly, apologies in the delay coming to this decision. Both reviewers have identified some minor (R1) and more significant (R2) issues with the manuscript. I would ask you to address especially the comments from reviewer 2 fully. I believe this can be carried out without need for further review (the paper has already had two sets of reviews) but will of course need to be checked by the editors so I encourage you to make the changes as fully as possible.

Re: We appreciate the Editor's assistance in finalizing the paper. We also sincerely value the thoughtful insights from both reviewers and acknowledge the issues that require revision. We have carefully addressed all comments and thoroughly revised the manuscript per the reviewers' suggestions.

Reviewer 2:

This is the review report for the manuscript, entitled: "Knickpoints and Fixpoints: The Evolution of Fluvial Morphology under the Combined Effect of Fault Uplift and Dam Obstruction on a Soft Bedrock River" by Chen et al. This study utilizes high-resolution elevation survey data combined with remote sensing images to obtain nearly 20 years of river longitudinal profiles with a resolution of submeter since 1999. The 1-D knickpoints evolution model based on the diffusion model, along with the digitization of sandbar within the river channel, contributes to the 3-D of river network configuration. Overall, the author successfully constrains the elevation changes of river channels and the main flow paths in three rivers along the Chelungpu Fault in central Taiwan. The author claims that the evolution of these three rivers can represent the influences of faults, dams, or a combination of both on river topography. This is highly meaningful research that distinguishes human activities from natural driving forces in shaping the natural landscape, providing insights into the field of anthropogenic geomorphology.

However, I share some concerns that need to be addressed before publication. Firstly, the quantification of contributing factors is crucial for establishing a comparative platform. The role of the fault: the vertical displacement caused by the 921 earthquake along the Chelungpu Fault led to a north-high and south-low displacement, creating discontinuities in river longitudinal profiles. Afterward, does the fault lose its role, and does it no longer contribute to tectonic uplift? Regarding the role of dams: how far can the tool effect of the dam persist, and does the nature of sediment on downstream change? The title may lead readers to expect a discussion of the similarities and differences between dams and knickpoints, but the content lacks corresponding discussion. Secondly, the manuscript extensively presents speculative inferences without citations. The author should avoid subjective inferences and excessive assumptions without literature support. The discussion section on the evolution processes of the three rivers is currently weak, primarily offering descriptive phenomena. Thirdly, restructuring sentences and simplifying images could enhance the readability of the paper.

Re: Thank you very much for your positive and detailed review of our manuscript. Most of the comments in the paragraph have been mentioned again in the following line-by-line comments. Most of the opinions have been responded to line by line below. Here, we will further address the impact of the earthquake and the basis of the inferences.

Firstly, according to paleoseismic studies, apart from the Chi-Chi earthquake, five paleo-earthquake events along the Chelungpu Fault over the past 2000 years can be roughly identified (1999, 1650-1520 AD, 1270-1160 AD, 1060-1030 AD, 570-400 AD,

and 240-50 AD) (Chen et al. 2004, 2005). This indicates that the likelihood of another large-scale earthquake causing surface uplift in the short term is low. The observed surface uplift was a one-time effect caused by the Chi-Chi earthquake in 1999. In the case of the Daan River, due to its soft rock geology, the uplifted terrain was eroded within just a few years, as explained in the text.

Furthermore, the tool effects of dam persist over time, but the degree of change varies depending on the strength of the geological formations. Significant erosion can form within a few years on soft rock, while on hard rock, this process could take several hundred years.

Our inferences are based on measured data, and in addition to using actual topographical changes as evidence, we also referenced the views of other researchers and mathematical models to validate our conclusions. Over the past 20 years, significant changes have been observed in three rivers within the study areas, which are strong evidence of the evolutions.

- Chen, H.C., Chang, and Y.H. Lee: Slip Rate and Recurrence Interval of the Chelungpu Fault During the Past 1900 Years: Quaternary International, 115-116, 167-176, 2004.

-Chen, W.S., Lee, K.J., Lee, L.S., Streig, A.R., Chang, H.C., Lin, C.W.: Significant sedimentation of coseismic fault-propagation growth-fold in the Chichi earthquake rupture, central Taiwan, accepted to Journal of Asian Earth Sciences, 2005.

Line-by-line comments:

1. L21 Are there significant differences in the impact of "natural" tectonic movements and "artificial" tectonic movements on river topography?

   Re: (We assume the reviewer is referring to artificial "structures.")

   Our response to this question is affirmative. This paper highlights the differences between the two, as both lead to distinct topographical changes. These changes can be likened to a moving point versus a fixed point on a riverbed, resulting in entirely different morphological evolution outcomes, as illustrated in Fig. 1(a) and 1(b).

2. L21 please specify "river equilibrium".

   Re: "River equilibrium" encompasses many aspects that represent a spatially and temporally stable state. Here, we are primarily referring to sediment transport, which affects riverbed morphology changes. We have slightly revised this sentence to improve clarity. L25 delete "significant". We have replaced the word with "substantial".

3. L25-L28 The formation of Knickpoints is repeated in these two sentences; a suggestion to rephrase is advisable.

   Re: The paragraph has been rephrased; please see lines25-29.

4. L30 delete "sudden"

Re: We replace it with "abrupt". It describes the sharpness and immediacy of the elevation change.(line32)

5. L30-31 The occurrence of knickpoints does not necessarily alter the slope of the river channel.

Re: The sentence in our manuscript is "The abrupt elevation change in the riverbed divides the river profile into two reaches with differing slopes…." Regarding the definition of a knickpoint, please refer to Li et al. (2021): "The knickpoint is usually defined as a sudden change in the slope of a river profile, and this change in slope is often placed in a power-law function to discuss its relationship with the drainage area. (Flint, 1974; Snyder et al., 2003; Wobus et al., 2006; Kirby and Whipple, 2012)"

- Li et al. (2021). Distribution and evolution of knickpoints along the Layue River, Eastern Himalayan Syntaxis. Journal of Hydrology, 603, 126915.

6. L32-37 please rephrase the text to better elucidate the theme of this paragraph.

Re: The paragraph has been rephrased; please see lines34-39.

7. L41-44 Information provided is overly fundamental.

Re: The sentences have been removed. Thank you for the suggestion.

8. L59-65 Suggest removing the introduction to the research area.

Re: We have shortened the sentences according to the suggestion.

9. Re: L71-72 Lacks appropriate citation of references.

The citations have been modified. (lines67-68)

10. L82 -83 Overly assertive.

Re: The sentence has been revised accordingly. (lines78-80)

11. L87-119 While the author has introduced the lithological settings of the riverbed, it is important to note that the incision or migration of the riverbed is also influenced by factors such as sediment particle size, sediment concentration, sediment distribution, and the compressive strength of rocks. Moreover, the discussion should ideally focus on the tectonic uplift during interseismic periods and the runoff discharge. Therefore, it is recommended that the author supplement relevant data to address these aspects in the study.

Re: We fully agree that migration of the riverbed is also influenced by those factors mentioned. We intend to present the most critical data. Fig. 2 and Fig. 13 reveal data on lithology and cumulative flow, respectively. More detailed information can be found in the *Open discussion*, and additional relevant data have been addressed in other research (e.g., Cook et al., 2013; Huang et al., 2013).

Considering that this paper's focus is not comparing sediment transport rates or

knickpoint migration speeds, and that has very little value merely listing data without further explanation and analysis, we prefer to maintain the current level of data presentation.

12. L121-133 The author has not appropriately separated the materials from the methods. There is a lack of clear explanation regarding the purpose of analyzing the width and depth of the river.

Re: We have modified the subtitle "materials" to "Data Collection and Analysis Methods" so the content in the subsection should be suitable now.

The width (W) and depth (D) of the river can be used to quantify changes in river patterns. In order to analyze the variation of channel width, depth, and aspect ratio (W/D), we calculated the bank-full discharge width and depth, representing the maximum flow that can occur in a river before water starts overflowing and spreading out onto the floodplain. (lines124-132)

13. L135-137 Reasons for the Need for Mathematical Models?

Re: The application of the mathematical model provides an abstract description of a concrete system using physical concepts and mathematical language. The results of the mathematical model can be validated against the actual topographical changes, demonstrating the evolution of the knickpoint.

14. L184-185 move to method

Re: The sentences have been moved.

15. Please maintain consistency in the name of the Jiji (Chichi) dam.

Re: The text consistently uses "Jiji dam." The label in Fig. 8 has been revised accordingly.

16. L324-331 Please provide information in the materials section regarding the source and processing of the flow data.

Re: The WRA provided the daily flow data, and we calculated the cumulative flow to compare the relationship between knickpoint retreat and discharge. (we have added the explanation in lines326-336)

17. Fig.1 Lacks information.

Re: We have added text on the sub-figure (c). Thanks for the comment.

18. The subfigure in Figure 5 is unclear and lacks sufficient information.

Re: Figure 5 primarily focuses on the actual terrain changes. The subfigure is used to illustrate the trends in terrain variation through a mathematical model and to show its similarity to the actual terrain. Therefore, we present the

mathematical model as a subfigure. We have revised the caption to clarify that the subfigure represents the results of the numerical model calculations.

19. Fig. 12a If the numbers on the T-bar represent the ratio of W/D, some values appear to be quite peculiar, for instance, the W/D values in the upper reaches of the Dajia River. I think a simple table can replace this figure.

    Re: Fig.12a illustrates the dramatic variation in W/D (width-to-depth ratio), and the results clearly show the deepening of the channel. We believe that when readers see the figure, they will distinctly sense the intensity of the channel deepening that we are aiming to highlight.

Review 1:

I have read through the revised manuscript and found that many of my concerns are well-addressed. I leave some additional comments regarding the organization of the manuscript and the influences of discharge and lithology on the evolution of the channel morphology.

I understand that the current study is built upon Cook et al. (2014) and agree that revisiting the results of Cook et al. (2014) helps understand the evolution of the channel morphology. Yet, I think the current descriptions regarding Daan River includes both results and interpretation and suggest reorganizing the corresponding sections. Or, maybe you can introduce Cook et al. (2014) in Study area to avoid the mixing of results and discussion.

Regarding your reply to the comment from reviewer 1 (the fourth one), I have found the argument on the effects of lithology convincing and interesting. I believe you can further strengthen the current manuscript by adding the same argument about the lithology.

Line by line comments:

1.  Line 199: I suggest citing Table 1 to show the consistency between the modeled and observed knickpoint retreat speed.

    Re: We did not find a suitable place to cite Table 1 around line 199, so we referenced it in line 205.

2.  Line 335: The sentence lacks a verb and looks incomplete.

    Re: The sentence was revised. (line326)

3.  Line 339: Suggest changing continuously to continuous and increasing to increase.

    Re: The sentence was revised according to the comment. (line330)

4.  Line 341-342: I could not understand what you meant by these two sentences. Since these sentences are the reply to the comment from reviewer1, I suppose you meant discharge variability does not affect the observed evolution of channel morphology. However, because discharge clearly dictates the knickpoint retreat speed when looking at the individual rivers, I wonder why the rates of knickpoint retreat are so different between the Dajia and Daan rivers. I do not think you need to find a clear answer to this question, but it is worth adding some sentences or a paragraph to the discussion.

    Re: The unclear sentences have been modified, and the paragraph now points out the difference in knickpoint retreat rates between the two rivers as the reviewer's comment. It would be hard to verify which reason is crucial, so we just provide some possible reasons. (lines332-336)

5.  Fig. 5: Maybe you should write that you used DSMs generated from aerial

photos in the body text, not just in the figure caption.

Re: We added the source of the DSMs in the section 2.2. (Lines 124-127). Thank you for the suggestion.

6. Fig.12: What do background thin-colored lines represent?

Re: Fig. 12(a) is a 3D diagram, where the lines serve as guides to help interpret the position of the "T" shape in the 3D space. The lines in Fig. 12(b) represent changes in the river facies, with the transition from lighter to darker lines depicting the variations in depth.

7. Fig. 13: Is the Y-axis label "Accumulated flow"? Also, since there is no knickpoint in Zhuoshui river, I wonder why Zhuoshui river is included.

Re: Thank you for the correction for the Y-axis. Indeed, there is no knickpoint in the Zhuoshui River; however, since this paper discusses the conditions of three rivers simultaneously, we believe that some readers might be interested in the discharge data of the Zhuoshui River, and we can easily provide that information to satisfy their curiosity.